



# Electrical Resistivity Dynamics beneath a Fractured Sedimentary Bedrock Riverbed in Response to Temperature and Groundwater/Surface Water Exchange

Colby M. Steelman[1], Celia S. Kennedy[2], Donovan Capes[2], Beth L. Parker[1,2]

[1]School of Engineering, University of Guelph

[2]School of Environmental Sciences, University of Guelph

*Correspondence to*: Colby M. Steelman (csteelma@uoguelph.ca)

**Abstract**. Bedrock rivers occur where surface water flows along an exposed rock surface. Fractured sedimentary
bedrock can exhibit variable groundwater residence times, anisotropic flow paths, heterogeneity, along with
diffusive exchange between fractures and rock matrix. These properties of the rock will affect thermal transients in
the riverbed and groundwater-surface water exchange. In this study, surface electrical methods were used as a non-
invasive technique to assess the scale and temporal variability of riverbed temperature and groundwater-surface
water exchange beneath a sedimentary bedrock riverbed. Conditions were monitored on a semi-daily to semi-
weekly interval over a full annual period that included a seasonal freeze-thaw cycle. Surface electromagnetic
induction and electrical resistivity imaging methods captured conditions beneath the riverbed along a pool-riffle
sequence within the Eramosa River, Guelph, Ontario, Canada. Geophysical datasets were accompanied by
continuous measurements of aqueous specific conductance, temperature and river stage. Vertical temperature
profiling conducted in an inclined borehole underlying the river revealed active groundwater flow zones through
fracture networks within the upper 10 m of rock. Resistivity measurements during cooler high-flow and warmer
low-flow conditions identified a spatiotemporal riverbed response that was largely dependent upon riverbed
morphology and groundwater temperature. Time-lapse resistivity profiles collected across the pool and riffle
identified seasonal transients within the upper 2 m and 3 m of rock, respectively, with spatial variations controlled
by riverbed morphology (pool verses riffle) and dominant surficial rock properties (competent verses weathered
rock rubble surface). While the pool and riffle both exhibited a dynamic resistivity through seasonal cooling and
warming cycles, conditions beneath the pool were more dynamic, largely due to the formation of river ice.
Although seasonal resistivity trends beneath the riverbed suggest groundwater discharge may be influencing the
spatiotemporal extent of a groundwater-surface water mixing zone, intraseasonal resistivity transience suggest
potential groundwater-surface water exchange across the upper few meters of rock.



## 1 Introduction

Fractured sedimentary bedrock represents an important source of water for many communities around the world. Although the effective porosity of rock is low relative to unconsolidated sediment, the existence of dense networks of interconnected fractures, dissolution-enhanced conduits, and karst features, can result in productive yet heterogeneous and anisotropic flow systems. An exposed bedrock surface may exhibit greater variability in flow and transport properties as it is subjected to weathering and erosional processes. This can lead to very complicated groundwater recharge and discharge patterns particularly in areas hosting dynamic interactions between groundwater and surface water.

Fractured rock may be conceptualized as a dual porosity system where fractures dominate flow, but remain connected to water stored in the porous matrix through advection and diffusion. Such conceptualizations of fracture flow and transport are routinely applied to groundwater resource (e.g., Novakowski and Lapcevic, 1998; Lemieux et al., 2009; Perrin et al., 2011) and contaminant transport studies (e.g., Zanini et al., 2000; Meyer et al., 2008; McLaren et al., 2012). While a number of recent studies have extended some of these concepts to fluvial depositional environments (e.g., Singha et al., 2008; Toran et al., 2013a), there remain gaps in our conceptual understanding of groundwater-surface water interaction and exchange mechanisms in bedrock rivers where discrete fracture networks will dominate groundwater-surface water flux with secondary interactions supported by the porous rock matrix.

Groundwater-surface water interactions at the reach-scale are conceptualized through gaining, losing and flow-through interactions (Woessner, 2000). At the channel scale, micro-to-macro bedform variations result in variably-scaled surface water downwelling (recharge) zones and groundwater upwelling (discharge) zones (e.g., Binley et al., 2013; Käser et al., 2013). Groundwater temperature measurements are routinely used to evaluate spatiotemporal variations in groundwater-surface water exchange or flux across riverbeds (e.g., Anderson, 2005; Irvine et al., 2016). Yet, very little is known about the existence and nature of groundwater-surface water mixing zones in fractured sedimentary bedrock, largely because these systems are very difficult to instrument using direct methods (e.g., drive point monitoring wells, seepage meters, thermistors) and the scale of the interaction may be very small or heterogeneous relative to equivalent processes in unconsolidated sediment.

Hydrologic processes along a fractured bedrock river were explored by Oxtobee and Novakowski (2002), who concluded that groundwater-surface water interaction was restricted by poor vertical connectivity and limited bedrock incision (i.e., exposure of bedding plane fractures). A subsequent numerical sensitivity analysis by Oxtobee and Novakowski (2003) confirmed that groundwater-surface water connectivity through discrete fractures would be highly variable in space and time, and would largely depend on fracture size or aperture, river stage, and the distribution of hydraulic head within the flow system. Fan et al. (2007) numerically explored the influence of larger-scale fracture orientations and geometries on the groundwater flow system near a stream. Although these previous studies offered valuable insights into the magnitude of groundwater-surface water exchange, they were based on idealized fracture network conceptualizations, and did not consider the role of matrix porosity and potential exchanges between fractures and the porous matrix.





Fractured sedimentary bedrock exhibits complex flow systems, where the bulk of the flow occurs in the fracture
network, with highly-variable head distributions; matrix storage may support equally complex biogeochemical
processes and thermal dynamics through convective or diffusive exchange with open fractures or dissolution-
enhanced features (Fig. 1). Ward et al. (2010a) demonstrated how surface electrical methods can be used to detect
and quantify diffusive mass transport (exchange) between a mobile and immobile storage zone in a shallow
riverbed. Therefore, we hypothesize that a groundwater-surface water mixing zone – encompassing fracture and
matrix flow and diffusion – may be identified within a fractured bedrock riverbed by monitoring spatiotemporal
changes groundwater temperature and porewater electrical conductivity using minimally invasive electrical
resistivity methods. The detection of seasonal transients beneath the bedrock riverbed would support future
conceptualizations of groundwater-surface exchange along fractured bedrock rivers.
Our study focuses on geoelectrical transients along the Eramosa River in Ontario, Canada. Seasonal variations in
electrical resistivity distribution were measured along two transects intersecting a pool-riffle sequence. Based on
continuous measurements of groundwater and surface water temperature, specific conductance and river stage,
spatiotemporal resistivity dynamics were largely controlled by riverbed morphology in combination with seasonal
changes in water temperature and electrolytic concentration. Over a complete annual cycle, formation of ground
frost and basal ice during the winter season was accompanied by stronger geoelectrical dynamics than intraseasonal
(spring, summer and fall) transience in the flow system. These geoelectrical observations support the existence of a
predominant groundwater discharge zone with limited groundwater-surface water mixing**.**
**2  Background**
**2.1  Geophysical Investigations along Streams and Rivers**
Electrical and electromagnetic methods such as ground-penetrating radar, electromagnetic induction and electrical
resistivity imaging are commonly used to characterize fluvial deposits (e.g., Naegeli et al., 1996; Gourry et al., 2003;
Froese et al., 2005; Sambuelli et al., 2007; Rucker et al., 2011; Orlando, 2013; Doro et al., 2013; Crosbie et al.,
2014). The capacity of time-lapse electrical resistivity imaging for conceptualization of groundwater transients in
sediment is also documented in the literature (e.g., Nyquist et al., 2008; Miller et al., 2008; Coscia et al., 2011;
Cardenas and Markowski, 2011; Musgrave and Binley, 2011; Coscia et al., 2012; Dimova et al., 2012; Wallin et al.,
2013). Electrical imaging of natural river systems perturbed by solute tracers has resulted in unprecedented
visualizations of fluid flow (e.g., Ward et al., 2010a, 2010b; Doetsch et al., 2012; Ward et al., 2012; Toran et al.,
2013a; Toran et al., 2013b; Harrington et al., 2014). More recent applications of electrical resistivity in karst
undergoing surface water transients have shown how surface geophysics can unravel complex hydrologic processes
in sedimentary bedrock environments (e.g., Meyerhoff et al., 2012; Meyerhoff et al., 2014; Sirieix et al., 2014),
especially when site conditions limit the use of direct measurement methods.
While a variety of geophysical tools and techniques can measure flow and water chemistry in space and time
(Singha et al., 2015), the most appropriate tool and approach will depend on the scale of interest. The vast majority
of geophysical work within shallow river environments has utilized discrete temperature monitoring below the





riverbed to assess vertical fluxes (e.g., White et al., 1987; Silliman and Booth, 1993; Evans et al., 1995; Alexander
and Caissie, 2003; Conant, 2004; Anderson, 2005; Hatch et al., 2006; Keery et al., 2007; Schmidt et al. 2007;
Constantz, 2008); these works have focused on processes within alluvial sediments. Recent advancements in
distributed fiber optic cables have improved spatial and temporal resolution of groundwater-surface water
interactions (e.g., Slater et al., 2010; Briggs et al., 2012; Johnson et al., 2012).
Groundwater and surface water interaction can be monitored through changes in thermal gradient or electrolytic
concentration (e.g., Norman and Cardenas, 2014), yet the scale and magnitude of these interactions will vary as a
function of riverbed architecture and subsurface hydraulic conditions (Crook et al., 2008; Boano et al., 2008; Ward
et al., 2012; Tinkler and Wohl, 1998) resulting in spatially dynamic exchange. These processes are further
complicated by diel (e.g., Swanson and Cardenas, 2010) and seasonal (e.g., Musgrave and Binley, 2011)
temperature fluctuations across a range of spatial scales, local transients such as precipitation events (e.g.,
Meyerhoff et al., 2012), river stage fluctuations (e.g., Bianchin et al., 2011) and controlled dam releases (e.g.,
Cardenas and Markowski, 2011). Relative to other non-invasive geophysical methods, electrical resistivity methods
are more robust in their ability to provide information about temperature and solute fluctuations beneath actively
flowing surface water bodies (e.g., Nyquist et al., 2008; Cardenas and Markowski, 2011; Ward et al., 2012;
Meyerhoff et al., 2014) particularly in a time-lapse manner. Unlike conventional hydrogeological methods (e.g.,
screened or open coreholes), which may bias conduction in the fractures, surface electrical methods are sensitive to
the bulk electrical conductivity of the formation, making them more suited for detection of processes between the
open fractures/conduits and the porous matrix.
**2.2 Electrical Properties of the Subsurface**
Electrical resistivity methods are based upon Ohm's Law ($R = \Delta V / I$). In the case of a homogeneous half-space,
the electrical resistance ($R$) of the subsurface is determined by measuring the potential difference ($\Delta V$) across a pair
of 'potential' electrodes due to an applied current ($I$) across a pair of 'current' electrodes some distance away. The
measured $R$ ($\Omega$) across a unit volume of the earth can be converted to apparent resistivity ($\Omega$ m) using a specific
geometric factor that compensates for varying electrode array geometry (Reynolds, 2012). Apparent resistivity
measurements are commonly interpreted using tomographic inversion techniques, whereby measured data is
reconstructed from forward models of an optimized physical parameter distribution (Snieder and Trampert, 1999;
Loke et al., 2013). Although data inversion techniques are standard practice in the interpretation of most
geophysical data, the model that best matches the measured data is not necessarily an exact representation of the
subsurface. The inversion process ultimately yields a smoothed representation of the actual parameter distribution.
The bulk electrical resistivity (i.e., inverse of conductivity) of a formation can be calculated through a simple
empirical relationship known as Archie's Law (Archie, 1942):
$$\rho_b = \phi^{-m}\rho_w, \qquad\qquad\qquad (1)$$
where the resistivity of the bulk formation ($\rho_b$) is simply related to the porosity of the medium ($\phi$) raised to the
negative power ($m$), which represents the degree of pore cementation, and the resistivity of the pore fluid ($\rho_w$)





(Glover 2010).  This relationship carries a number of simplifying assumptions: the most significant being that the
current flow is entirely electrolytic.  While more sophisticated formulations of Archie's Law incorporating fluid
saturation and interfacial conduction can be found in the literature (e.g., Rhoades et al., 1976; Waxman and Smits,
1968), Eq. (1) is considered to be a reasonable approximation in a saturated relatively clay-free environment.
Equation (1) is used in this study to evaluate the impact of observed groundwater and surface water aqueous
conductivity variations on the bulk formation resistivity.  Here, a value of 1.4 was used for the constant $m$, which is
considered reasonable for fractured dolostone.  It should be noted that the relative impact of aqueous conductivity
changes on the bulk formation resistivity may vary with clay content and pore connectivity due to intrinsic
deviations in the $m$ value (Worthington, 1993).  Furthermore, orientated fracture networks may result in an
anisotropic resistivity response (Steelman et al., 2015b); however, these static properties of the rock will not impact
relative changes in resistivity at a fixed location.
The electrolytic (fluid phase) resistivity will depend on the concentration and composition of dissolved ions, and
viscosity of the pore water (Knight and Endres, 2005).  Increasing ion concentrations and temperature will lead to a
reduction in formation resistivity.  Empirical evidence has shown that resistivity can decrease anywhere from 1 % to
2.5 % per °C (Campbell et al., 1948; Keller, 1989; Brassington, 1998).  Temperature corrections can be made using
Arp's law (Arps, 1953):
$$\rho_{w2} = \frac{\rho_{w1}(T_1 + 21.5)}{(T_2 + 21.5)},$$    (2)
where $\rho_w$ ($\Omega$ m) and $T$ (°C) represent the resistivity and temperature of the water at two points.  This formulation
was developed from a least-squares fit to the conductivity of a NaCl solution ranging from 0°C to 156°C; however,
the exact relationship between fluid conductivity and temperature will depend on the composition of the electrolytic
solution (Ellis, 1987).
**2.3  Field Site Description**
The Eramosa River is a major tributary of the Speed River within the Grand River Watershed, Ontario, Canada, and
resides upon a bedrock aquifer of densely fractured dolostone of Silurian age with dissolution-enhanced conduits
and karst features (e.g., Kunert et al., 1998; Kunert and Coniglio, 2002; Cole et al., 2009).  Outcrops, core logs and
geophysical data collected along the Eramosa River (e.g., Steelman et al., 2015a; Steelman et al., 2015b) indicate
abundant vertical and horizontal fracture networks and karst features intersecting and underling the river.  These
field observations support the existence of a potential groundwater-surface water mixing controlled by discrete
fracture networks and dissolution-enhanced features.
A focused geophysical investigation was carried out along a 200 m reach of the Eramosa River (Fig. 2).  The study
area was positioned at a bend in the river with relatively cleared vegetation along the south shoreline and adjacent
floodplain with exposed rock at surface.  A network of coreholes (continuously cored boreholes) and streambed
piezometers were installed across the site.  Locally, the water table elevation corresponds to the surface water or
stage elevation, resulting in vadose zone thicknesses between <0.5 m to 2.0 m along the shorelines.  The temperate





southern Ontario climate subjects the river to a wide-range of seasonal conditions, including high precipitation
periods in spring and fall, hot and dry summers, and variable degrees of ground frost and surface water freeze-up
during the winter months (Fig. 3a).

Locally, the river incises the Eramosa Formation by 2 m to 3 m exposing abundant vertical and horizontal fractures
with little to no alluvial sediment deposited along the riverbed (Fig. 3b). Regionally the Eramosa acts as a
discontinuous aquitard unit (Cole et al. 2009); however, core logs collected at the study site show bedding plane and
vertical joint set fractures spanning the entire 11 m sequence of Eramosa. This upper formation is underlain by
approximately 3 m of cherty, marble-like Goat Island formation, which exhibits high-angle fractures along cherty
nodules near the Eramosa contact. The Goat Island is underlain by more than 15 m of Gasport, which exhibits coral
reef mounds of variable morphology. The rock matrix of the Gasport is visually more porous with well-defined
vugs, dissolution-enhanced features, and fewer fractures than the overlying Goat Island and Eramosa. A full-
description of these bedrock sequences can be found in Brunton (2009).

In this region, the winter season may be accompanied by ground frost formation and variable surface water freezing.
Seasonal freeze-up will consist of an ice crust layer on the surface of the water and the possible formation of basal
ice along the riverbed (Stickler and Alfredsen, 2009). This phenomenon can occur during extreme atmospheric
cooling over turbulent water bodies, resulting in super-cooled water (<0°C) that rapidly crystalizes to form frazil
(i.e., tiny ice particles with adhesive characteristics); these crystals can flocculate to form slush, which adheres and
accumulates on the substrate forming a basal ice layer.

## 3 Methods

### 3.1 Bedrock Lithology, Fractures and Porosity

The geology was characterized through a series of vertical and angled coreholes along the southern shoreline that
were advanced into upper Gasport formation. These drilling activities were part of a broader hydrogeological
investigation of groundwater flow and fluxes along the river. A network of riverbed piezometers, bedrock stage
gauges, and flux measurement devices were installed between 2013 and 2014 within the pool. Locally, the riverbed
morphology can be distinguished in terms of the amount of bedrock rubble or weathered rock fragments blanketing
the exposed rock surface. Figure 2 shows the transition from a rubble dominated riverbed (RDR) to a more
competent rock riverbed (CRR); this boundary roughly corresponds to the riffle-pool transition.

Geophysical measurements were supported by temperature, specific conductance of the fluid and river stage
elevation collected at nearby monitoring points (Fig. 2). The geologic and hydrogeologic data were obtained from a
river stage gauge (RSG4), a vertical corehole (SCV6) drilled to a depth of 10.9 m, and an angled corehole (SCA1)
drilled to a vertical depth of 31.8 m. The angled corehole plunges at 60° and is orientated at 340°, and therefore,
plunges beneath the river with a lateral footprint spanning approximately 21 m from its surface expression.
Coreholes were drilled using a small-diameter portable Hydrocore Prospector™ drill with a diamond bit (NQ size:
47.6 mm core and 75.7 mm corehole diameter) and completed with steel casings set into concrete to a depth of 0.6



m below ground surface (bgs). All coreholes were sealed using a flexible impermeable liner filled with river water
(FLUTe™ Flexible Liner Underground Technologies, Alcalde, New Mexico, USA) (Keller et al., 2014).
The SCA1 rock core was logged for changes in lithology, vugs, and fracture characteristics, intensity and
orientation, including bedding plane partings. Rock core subsamples were extracted for laboratory measurements of
matrix porosity using the following procedure: sample was oven dried at 40°C; dimensions and dry mass recorded;
samples evacuated in a sealed chamber and imbibed with deionized water; sample chamber pressurized to 200 psi to
300 psi for 15 minutes; samples blotted and weighed to obtain saturated mass. Open coreholes were logged using an
acoustic (QL40-ABI) and an optical (QL40-OBI) borehole imager (Advanced Logic Technologies, Redange,
Luxembourg), to characterize the fracture network.
**3.2 Pressure, Temperature, Specific Conductance and River Flux**
Temperature, specific conductance and hydraulic pressure data were recorded using a CTD-Diver™ (Van Essen
Instruments, Kitchener, Canada) deployed within RSG4 (surface water) and SCV6 (groundwater) at a depth of 10.5
m bgs. The transducer in SCV6 was placed near the bottom of the open corehole prior to being sealed with an
impermeable liner, thereby creating a depth-discrete groundwater monitoring point. Surface water data were
recorded through the full study period while deeper bedrock conditions were recorded from early-September 2014
through late-May 2015. All measurements were collected at 15 minute intervals.
Vertical temperature profiles were additionally collected along the inclined sealed corehole water column of SCA1
from 4-Sep-2014 to 22-May-2015 using an RBR*solo*™ temperature logger paired with a RBR*solo*™ pressure logger
(RBR Limited, Ottawa, Canada). These data were recorded at 0.5 second intervals while the sensors were manually
lowered into the water column using a fiberglass measuring tape at a rate of 0.02 m s$^{-1}$ to 0.03 m s$^{-1}$. Barometric
pressure was collected at the site using a Baro-Diver™ (Van Essen Instruments, Kitchener, Canada).
Rainfall was recorded at the University of Guelph Turfgrass Institute, located 2 km northwest of the site, while
snowfall accumulation was obtained from the Region of Waterloo Airport roughly 18 km south west of the site.
Hourly mean river flux was recorded 900 m upstream at the Watson Road gauge operated by the Grand River
Conservation Authority. A summary of the weather and river flux data are provided in Fig. 4.
**3.3 Riverbed Electrical Resistivity**
**3.3.1 Spatial Electrical Resistivity Mapping**
Riverbed electrical resistivity distribution was initially measured using a Geonics EM-31 ground conductivity meter
(Geonics, Mississauga, Canada) during a seasonally cool and warm period: early-spring/high-stage conditions on 3-
Apr-2013 and mid-summer/low-stage conditions on 7-Jul-2014. Measurements were collected at a rate of 3
readings per second with the device operated in vertical dipole mode held ~1 m above the riverbed. The effective
sensing depth of this instrument in vertical dipole mode is approximately 6 m, and is minimally sensitive to
conditions above the ground surface (McNeil, 1980). Data was recorded along roughly parallel lines spaced ~1.75
m apart orthogonal to the river orientation, with the coils aligned parallel to surface water flow direction. Water





depths over the investigated reach varied from <0.1 m in the riffle during low-flow to nearly 1 m in the pool during
high-flow conditions.  Data sets were filtered for anomalous outliers prior to minimum curvature gridding.

### 3.3.2    Time-Lapse Electrical Resistivity Imaging

Surface electrical resistivity measurements were collected along two transects orientated orthogonal to the river (Fig.
2), capturing conditions within a pool and riffle sequence (Fig. 3).  Line 1 was positioned downstream over a deeper
pool section with more substantial bedrock incision into a competent bedrock surface (Fig. 3b, i and ii), while line 2
was situated upstream over a shallower riffle section blanked by bedrock rubble fragments with less bedrock
incision (Fig. 3b, iii).
For this study, resistivity cables were constructed using a pair of 25 multicore cables (22 gauge strained wire, 600V
rating) wound within a PVC jacket.  The PVC jacket was split open every meter to expose and cut out a single wire
that was connected to an audio-style banana plug.  Spliced sections of outer PVC jacket were resealed using heat
shrink tubing and silicon.  This process resulted in two 24 channel cables each connected to a single multi-pin
connector for direct data logger communication.  Electrodes were constructed from half-inch diameter stainless steel
rod cut to 6 inch lengths.  A hole was drilled on one end of the electrode to receive the banana plug connector.
Given the exposed bedrock across the site, a half-inch hole was drilled into the rock at 1 m intervals along the
ground surface.  In some cases, electrodes were buried beneath a rubble zone of the riverbed, or were pushed into a
thin layer of sediment.  On the shorelines electrodes were fully implanted into the rock along with a few teaspoons
of bentonite clay to minimize contact resistance.  Each monitoring line was instrumented with dedicated electrodes
and cables that remained in place for the duration of the study.
Resistivity measurements were recorded using a Syscal Junior Switch 48 (Iris Instruments, Orléans, France)
resistivity meter.  A Wenner array was selected for its higher S/N ratio.  A dipole-dipole array was tested, but found
to be very susceptible to noise (i.e., excessive number of bad data points due to low measured potentials); this was
attributed to the high-contact resistances with rock combined with the instruments moderate power capability (max
400 V, 1.3 A).  Although the Wenner array geometry results in a stronger signal (i.e., potentials are measured across
a pair of electrodes located between the current electrodes with an equal inter-electrode spacing), it will be less
sensitive to lateral variations across the riverbed compared to the dipole-dipole array, and thus, less sensitive to the
presence of a single or package of vertical fractures between adjacent electrodes.  Surface resistivity data were
recorded on a semi-daily to semi-weekly interval from 18-Jul-2014 to 3-Jul-2015 covering a complete annual cycle,
which included a seasonal freeze-thaw cycle, and numerous wetting-drying events accompanied by large river stage
fluctuations.  The timing of resistivity measurement events are shown with corresponding river flow rates and
atmospheric data in Fig. 4.  Resistivity measurements were generally recorded between 8 AM and 1 PM.
Measured apparent resistivity data was manually filtered to remove erroneous data points prior to being inverted
using RES2DINV v.3.59 (Geotomo Software, Malaysia), which uses the Gauss-Newton least-squares method (Loke
and Dahlin, 2002).  For this study, a robust inversion scheme was used with moderate to high dampening factors
given the high resistivity contrast observed along the surface, and intermittently noisy datasets.  The width of the





model cells were set to half the electrode spacing (i.e., model refinement) to help supress the effects of large surface
resistivity variations on the inversion process. All other parameters within the program were optimized to
compensate for high noise and large resistivity contrasts while achieving the lowest possible model root mean
squared (RMS) error.
Figure 5 shows the model setup for the pool and riffle, including the minimum and maximum river stage elevations
observed during the geophysical monitoring events. A portion of the electrodes were variably submerged beneath a
surface water layer. Stage elevations ranged from 310.92 masl to 311.32 masl at line 1, and 311.09 masl to 311.48
masl at line 2. Thus, each model was independently inverted with a defined surface water boundary (i.e, stage
height above the submerged electrodes) and true aqueous resistivity, both of which were fixed for each model
inversion. Model convergence typically occurred within 8 iterations.
Temporal variations in bedrock resistivity were assessed within four representative zones (A, B, C and D; Fig. 5)
using a resistivity index ($RI$). These zones were chosen based on their contrasting bedrock conditions, relative
position along the river transect and geophysical dynamics observed during the monitoring period. The $RI$ was
calculated for the pool and riffle resistivity profile as follows:
$$RI_{i,j} = \frac{MZR_{i,j} - MAR}{MAR},$$
(3)

where $RI_{i,j}$ = resistivity index for the i$^{th}$ zone on the j$^{th}$ sample date; $MZR_{i,j}$ = mean zone resistivity for the i$^{th}$ zone
on the j$^{th}$ sample date; $MAR$ = mean annual resistivity of the entire profile across the full time series for the pool or
riffle.
**4 Results**
**4.1 Bedrock Fracture Network, Temperature and Specific Conductance**
Formation contacts of the Eramosa–Goat Island and the Goat Island–Gasport formations were identified in core at
depths of 8.6 and 13.0 m bgs, respectively (Fig. 6a). Fractures beneath the river were predominantly horizontal to
slightly dipping (<10°), and most abundant in the Eramosa and Goat Island. Although vertical and sub-vertical
fractures (>10°) were relatively less abundant, they were more uniformly distributed with depth. These high-angle
fractures terminate at surface as vertical joint sets along two regional orientations: 10° to 20° NNE and 280° to 290°
SNW (Fig. 3b, ii). Matrix porosities from the corehole were relatively low, ranging from 0.5 % to 5 %, with the
lowest porosities observed along the highly weathered riverbed surface and lower portion of the Eramosa Formation.
Hydraulic head data collected in the river and at the base of SCV6 (10.5 m bgs) suggest a seasonally sustained
upward vertical gradient (i.e., groundwater discharge zone) at the pool.
Vertical temperature profiling within the static water column of the FLUTe™ lined SCA1 corehole from 4-Sep-
2014 to 22-May-2015 captured seasonal fluctuations in ambient groundwater temperature to depths up to 20 m (Fig.
6b), thereby delineating the vertical extent of the heterothermic zone. Temperatures inside the liner ranged from



18°C in late-summer, to 5°C in mid-winter.  Although fluctuations were observed along the entire 20 m profile, the
bulk of the variations (short and long-period) were observed in the upper 10 m bgs.
Previous studies using ambient temperature profiling in lined coreholes (Pehme et al. 2010; Pehme et al. 2014)
examined the effects of active groundwater flow around static water columns.  Pehme et al. (2010) demonstrated
how a lined water-filled corehole in thermal disequilibrium with the surrounding formation would exhibit more
short-period temperature perturbations along its vertical profile than an equilibrated water column within zones of
active groundwater flow.  Here, the onset of winter seasonal conditions (9-Jan-2015 through 31-Mar-2015) cooled
the corehole water column near the ground surface resulting in density-driven convection within the column, leading
to thermal disequilibrium with respect to the surrounding bedrock resulting in abrupt temperature perturbations as
the water column cooled toward 5°C.  The magnitude and frequency of the perturbations observed in Fig. 6b during
these cooler periods correspond to areas of increased fracture frequency (Fig. 6a), indicating active groundwater
flow zones beneath the riverbed.
Specific conductance and temperature of surface water (RSG4) and groundwater (SCV6) corresponding to
geophysical sampling events (Fig. 4) are presented in Fig. 7.  These data indicate that surface water specific
conductivity varied within a much narrower range than the actual (uncompensated) conductivity, which includes the
effects of temperature.  While the overall impact of temperature and ionic concentration on the specific conductance
of surface water were similar (i.e., equivalently dynamic), variations associated with ionic concentration appear
more erratic, and exhibited sharper fluctuations over shorter periods of time.  For instance, major precipitation
events coinciding with measurement events 13, 26 and 31 (refer to Fig.4) were accompanied by short-period
reductions in surface water conductivity and increases in temperature.  Seasonal atmospheric temperature trends
resulted in more gradual, yet seasonally sustained reductions in aqueous conductivity.  In comparison, the
groundwater specific conductance at 10.5 m bgs was comparatively stable during the study period, exhibiting a
moderate temperature driven decline superimposed by shorter-period fluctuations associated with ion concentration.
Figure 8 shows the potential impact of these observed specific conductivity and temperature variations (based on
Fig. 6b and 7) on the bulk formation resistivity using Eq. (1) and Eq. (2) for three representative porosity values.
Porosities of 1 % and 5 % correspond to the values obtained in core, while a porosity of 35 % might represent the
maximum porosity of a weathered or broken rubble zone.  These calculations indicate that variations in temperature
will likely be the primary driver in formation resistivity dynamics.  For instance, water temperature could affect the
formation resistivity by as much as 46 %, based on the observed range in groundwater and surface water
temperatures, respectively.  In comparison, measured aqueous conductivity ranges (along a particular isotherm) for
groundwater and surface water would affect the formation resistivity by 18 % and 36 %, respectively.  These
maximum effects represent end-member conditions for a specific porosity.  The natural system will exhibit a much
more complex distribution of formation resistivity given variable fracture networks, matrix porosity, and
dissolution-enhanced features.
**4.2  Sub-Riverbed Electrical Resistivity Distribution**





Two ground conductivity surveys were conducted across the riverbed to assess spatial variability in bulk formation
resistivity and its relationship to riverbed morphology (e.g., pool vs. riffle): the first resistivity snapshot was
collected on 3-Apr-2013 during high-flow conditions (6.81 $m^3 s^{-1}$) (Fig. 9a) while the second was collected on 7-Jul-
2014 during low-flow conditions (1.30 $m^3 s^{-1}$) (Fig. 9b). The daily average river flows for the years 2013 and 2014
were 3.5 $m^3 s^{-1}$ and 3.3 $m^3 s^{-1}$, respectively.
Two main observations can be made from the changes observed between cooler high-flow and warmer low-flow
conditions. First, the southern shoreline exhibited the highest resistivity (red areas) with the least temporal
variability. These areas are characterized by more competent and less-fractured rock (Fig. 3b, i). Secondly, a more
dynamic response was observed northward into the thalweg and along the north shoreline; the rock surface in these
areas was more weathered with large irregular rock fragments and dissolution features. A lower resistivity zone
(blue area) was identified upstream within the northern portion of the riffle section (Fig. 3b, iii). The riffle portion
of the river was also accompanied by a break in the high resistivity trend observed along the south shoreline. A
lower average resistivity was observed during warmer low-flow conditions indicating that a portion of the response
may be dependent on formation temperature (i.e., 5°C to 20°C fluctuations). Formation resistivities varied up to 10
% within the pool and up to 18 % within the riffle. While the average change in riverbed resistivity was 16 %,
portions of the riverbed further down and upstream of the resistivity transects did exhibit early-spring to mid-
summer fluctuations up to 30 %.

### 350 4.3 Time-Lapse Electrical Resistivity Imaging

### 351 4.3.1 Electrical Resistivity Models

Figure 10 provides a summary of the inverted model results at the pool and riffle sections for the full study period.
The mean inverted model resistivity and data range for each sample event is presented along with the number of
apparent resistivity data points removed from the dataset prior to inversion, and the root mean squared (RMS) error
of the inverted model. Although the resistivity distribution across the pool remained systematically higher than the
riffle throughout the entire monitoring period, both locations exhibited a dynamic response over the annual cycle. A
greater number of measurements had to be removed prior to inversion of frozen-period data sets, which may have
contributed to the higher RMS errors encountered during the winter period. A subset of the inverted resistivity
models over the annual cycle (i.e., samples a–h identified in Fig. 10) are shown in Fig. 11. These snapshots capture
the spatiotemporal evolution of predominant geoelectrical conditions beneath the riverbed.
Spatial electrical resistivity data were highly variable across the pool (Fig. 11a–h). The highest resistivities were
observed along the south shoreline, which coincided with the presence of competent bedrock (Fig. 3b, i), with
limited vertical and horizontal fractures. Similarly resistive conditions extended southward onto the floodplain.
Subsurface conditions became less resistive toward the north shoreline, which coincided with the presence of
increased fractures and dissolution features, mechanically broken or weathered bedrock, and a thin layer of organic
rich sediment alongside the north shoreline and floodplain. Initial surveys conducted across the pool on 25-Jul-2014
identified a relatively low resistivity zone (<1000 Ω m) extending 2 m beneath the riverbed that spanned the full





width of the river. Measurements on 26-Sep-2014 through 24-Dec-2014 captured the retraction of this zone toward
the north shore. During this period the resistivity across the full transect increased only slightly. The onset of
frozen ground and river conditions on 29-Jan-2014 resulted in an abrupt shift in the resistivity distribution. A high
resistivity zone formed above the water table across the southern floodplain and was accompanied by an increase in
resistivity across the full river profile. It is important to note that these frozen periods were accompanied by higher
model RMS errors, and thus, our interpretation of these data focus on long-period trends. The formation of river ice
(basal and surface ice) may have altered the true geometry of the surface water body represented in the model,
potentially contributing to the higher RMS errors. The arrival of seasonal thaw conditions on 27-Mar-2015 was
accompanied by reduced resistivities across the river as rock and river ice progressively thawed and was mobilized
by spring freshet. Further seasonal warming on 6-May-2015 and 3-July-2015 resulted in a systematic decrease in
riverbed resistivity from the north to south shoreline.
Riverbed resistivity across the riffle portion of the river (Fig. 11a–h) was markedly different with respect to the
distribution and magnitude of resistivity fluctuations. The riffle exhibited a zone of comparatively low resistivity
($<100$ $\Omega$ m) that extended slightly deeper than that at the pool, to a depth of 3 m. The initial survey on 26-Jul-2014
identified a zone of very low resistivity that progressively became more resistive over time (26-Sep-2014 through
24-Dec-2014). Much like the pool, however, this low resistivity zone reverted back toward the north shoreline. The
onset of seasonally frozen river conditions was accompanied by an increase in resistivity across a significant portion
of the riverbed. Inverse models during frozen water conditions were again accompanied by higher RMS errors,
which we attribute to the formation of river ice which could not be accounted for in the model. Unlike the pool,
which experienced the formation of a substantial zone of ground frost along the south shore, less ground frost was
observed at depth along the riverbanks bounding the riffle. Spring thaw brought reduced resistivities across the
riverbed with subtle lateral variations, followed by the beginnings of a less-resistive riverbed zone emanating
southward from the north shoreline. The bedrock resistivity below 3 m depth remained relatively constant through
the monitoring period.

### 4.3.2 Spatiotemporal Resistivity Trends

A resistivity index (RI) was calculated using Eq. (3) to assess spatiotemporal variations in electrical resistivity
within predefined zones (Fig. 5) of the bedrock beneath the river (Fig. 12); zones A and D represent conditions
along the south and north riverbank, while zones B and C represent conditions within the southern and northern
portions the river. These zones were defined based on their representative areas and the magnitude of the temporal
fluctuations observed over the full monitoring period (Fig. 11). A RI of zero indicates a mean zone resistivity
(MZR) that is equal to the mean annual resistivity (MAR) of the whole profile. An index of +1 indicates a
resistivity that is twice the annual mean, while an index of -0.5 indicates a resistivity that is half the annual mean.
The RI time-series for the pool (Fig. 12a) and riffle (Fig. 12b) capture the magnitude and frequency of the temporal
variability observed within these four representative zones. Relative to the MAR, the pool exhibited larger and more
frequent fluctuations in resistivity compared to the riffle. The south shoreline (zone A) at the pool was more
dynamic than the corresponding zone at the riffle; zone A at the pool encompasses a larger unsaturated zone, which



is more likely impacted by changes in temperature and saturation, especially during the freezing and thawing period.
The north shoreline (zone D) at the pool and riffle exhibited lower than average resistivities with relatively minimal
transience over the study period, with the exception of the mid-to-late-winter freeze-up. Here, a variable layer of
sediment and organic matter with higher water content likely moderated freeze-thaw fluctuations relative to sections
of exposed rock. Conditions below the riverbed (zones B and C) exhibited both longer-period (seasonal) and
shorter-period (intraseasonal) fluctuations. While the relative changes observed at the pool were larger than the
riffle, similar seasonal trends were observed at each location. Zones B and C at the pool were mutually consistent,
while those at the riffle were somewhat less consistent.
Although perturbations were observed in the resistivity beneath the riverbed before and after winter freeze-up (e.g.,
zones B and C), the responses were significantly dampened relative to the winter period. Events 13, 26 and 31 (Fig.
4 and 7), which correspond to periods of increase precipitation, may coincide with observed perturbations in the RI;
however, based on these data it is not clear whether the riverbed resistivity and surface water responses are mutually
consistent; this limited correlation may suggest that groundwater discharge in this section of the river is strong, and
thus, limiting potential groundwater-surface water mixing, at least at the spatial scales considered in this study.
Therefore, these observed geophysical dynamics within the riverbed may be associated with seasonal temperature
transience with secondary effects due to solute fluctuations.
**5.0  Discussion**
**5.1  Influence of Water Properties on Formation Resistivity**
Riverbed electrical resistivity mapping during high and low stage conditions identified a spatiotemporal response
within the upper 6 m of rock, which varied with riverbed morphology. Long-term resistivity monitoring along fixed
profiles over the pool and riffle portions of the river revealed a transient zone within the upper 2 and 3 m of bedrock,
respectively. In particular, the formation of a low resistivity zone (high electrical conductivity) was observed during
the warmer summer period that diminished as the environment cooled. While pore water conductivity depends on
electrolytic concentration and temperature, their individual contribution to the bulk formation response cannot be
decoupled across the entire study area given inherent/practical limitations in the number of direct measurement
points in a bedrock environment. Although this ambiguity in the driving mechanism of observed electrical changes
below the riverbed hinders our ability to definitively define the vertical extent of a potential groundwater-surface
water mixing zone, our geophysical data set does suggest that a groundwater-surface water mixing in a bedrock
environment may be limited, in part by strong upward hydraulic gradients and groundwater discharge at this site.
Aqueous temperature and specific conductance measurements collected in the river stage gauge (RSG1) and shallow
bedrock well (SCV6) provided end-member conditions (Fig. 7). These data were used to assess the influence of
aqueous conditions on bulk formation resistivity (Fig. 8). While some degree of overlap was observed between
groundwater and surface water properties, they were generally differentiable across the study area. That said,
aqueous temperature fluctuations likely dominated the bulk electrical response over the full annual cycle. Given the
impact of temperature on the bulk formation resistivity, observed bedrock resistivity dynamics are attributed to





changes in water/rock temperature with secondary effects caused by changes in electrolytic concentration. These
findings are consistent with Musgrave and Binley (2011), who concluded that temperature fluctuations over an
annual cycle within a temperate wetland environment with groundwater electrical conductivities ranging from 400
$\mu S\ cm^{-1}$ to 850 $\mu S\ cm^{-1}$ dominated formation resistivity transience. Of course, our annual temperature range was
more extreme than that of Musgrave and Binley, we examined electrical dynamics within a very different medium
(rock verses organic rich sediment), and ultimately captured a broader range of seasonal conditions that included
ground frost and riverbed ice formation.
Measurements collected with a shorter time-step (diurnal) and shorter electrode spacing may capture more transient
rainfall or snowpack melt episodes, possibly leading to the identification of electrolytic-induced transients beneath
the riverbed indicative of a groundwater-surface water mixing zone. Based on the short period of intraseasonal
fluctuations observed in Fig. 12, and the timing and duration of major precipitation or thawing events (5 to 7 day
cycles) (Fig.4), it is reasonable to assume that our geophysical time step (days to weeks) was accompanied by some
degree of aliasing. Finally, it is possible that shallower sections of rock within the river exposed to direct sunlight
during the day, which can vary depending on cloud cover (daily) and the suns position in the sky (seasonally), may
have exhibited a wider range, or more transient temperature fluctuations, than those areas beneath or adjacent to a
canopy. A closer inspection of the unfrozen temporal response in zone B reveals a wider range in resistivity relative
to the more northern zone C. At this latitude in the northern hemisphere the south shore will receive more direct
sunlight; it is possible that the shallow rock on this side of the river experienced greater fluctuations in temperature
(both seasonally and diurnally), thereby contributing to the observed geoelectrical dynamics.

### 5.2  Formation of Ground Frost and Anchor Ice

A dramatic increase in bedrock resistivity was observed with the onset of freezing ground conditions; this can
impact a wide range of infrastructure (e.g., dams, hydropower generation), ecologic (e.g., alteration of fish and
benthic habitats) and hydraulic functions (e.g., river storage, baseflow) (Beltaos and Burrell 2015). The formation of
a highly resistivity zone consistent with a seasonal frost front within the unsaturated portion of the riverbank (Fig.
11e; zone A in Fig. 12a), and the accumulation of river ice resulted in marked changes in resistivity. These winter
season effects are readily evident in Fig. 12. The magnitude of the resistivity increase observed at the pool and riffle
may suggest a potential reduction in the hydraulic connectivity between surface water and groundwater during the
winter months.
Ground frost primarily formed along the riverbank over the southern floodplain at the pool. This topographically
higher area was relatively devoid of large vegetation (Fig. 2), and thus, experienced more severe weather conditions
(e.g., higher winds resulted in less snow pack to insulate the ground). These conditions likely enhanced the
formation of a thick frost zone which propagated to the water table (pool in Fig. 11e). The adjacent northern
riverbank and those up-gradient at the riffle were topographically lower (i.e., thinner unsaturated zone) and were
sheltered by large trees. The formation of a seasonal frost zone along the riverbank may have implications to
baseflow dynamics during the winter months and early-thaw period.





The presence of river ice did have a noticeable impact on the inverse model results as reflected in the higher RMS
errors at the pool (>6 %) and riffle (>4 %) during the winter period (Fig. 10). This was particularly evident at the
pool, which exhibited a more uniform high resistivity zone beneath the riverbed (Fig. 11e). A simple sensitivity
analysis of the inversion process using different constraints on surface water geometry and aqueous electrical
resistivity suggests that model convergence was highly dependent on surface water geometry and its actual
resistivity. For instance, applying an aqueous resistivity of one-half the true value led to very poor model
convergence and substantial overestimates of river resistivity. The riffle was relatively less sensitive to surface
water properties likely because of its overall lower river stage compared to the pool, and hence, relatively lower
impact of the surface water body on the apparent resistivity measurement. This sensitivity to surface water
properties is a consequence of the high electrical contrast between the conductive surface water and resistive
bedrock. Anchor ice further reduced the electrical connectivity across the riverbed, while the ice crust along the
surface of the water altered the effective geometry of the water body, thereby impacting the inverse solution.
Unfortunately, direct measurements of river ice thickness were not collected, and thus could not be explicitly
incorporated into the surface water layer conceptualization during the winter months. Nevertheless, the effect of
river ice on the inverse models appears to be limited to the rock immediately beneath the surface water layer.
**5.3 Implications to the Conceptualization of Groundwater-Surface Water Exchange in Bedrock Rivers**
The fractured dolostone in this study consists of a visible orthogonal joint network approximately oriented at 10°
to 20° NNE and 280° to 290° SNW, consistent with the regional joint orientations, with frequencies ranging from
centimeter to sub-meter scale where exposed at surface. Streambed resistivity measurements indicate a dynamic
groundwater zone within the upper 2 m to 3 m of riverbed. A less-resistive zone (<1000 $\Omega$ m) was observed
beneath the pool emanating from the north shoreline during warmer low-flow periods (July and August 2014). This
zone retracted in late-summer but showed signs of reappearance in early July 2015. A similarly evolving low-
resistivity zone (<100 $\Omega$ m) was observed across the riffle, but was more variable across the river transect. While
dynamic fluctuations in temperature and aqueous conductivity support the potential existence of a definitive
groundwater-surface water mixing zone, it is not yet clear how these geoelectrical dynamics were influenced or
enhanced by fluid flow in the discrete fractures, or seasonal thermal gradients across the riverbed.
While groundwater-surface water exchange within a fractured bedrock river are expected, discrete fracture networks
and dissolution-enhanced features will support more heterogeneous and anisotropic surface water mixing zones
compared to porous unconsolidated sediment. Swanson and Cardenas (2010) examined the utility of using heat as a
tracer of groundwater-surface water exchange across a pool-riffle-pool sequence. Observed thermal patterns and
zones of influence (i.e., effective mixing zones) in their study were consistent with conceptual models depicting a
pool-riffle-pool sequence. While similar temperature dynamics may be expected across the pool-riffle-pool
sequence in a bedrock environment, our coarser temporal sampling interval (days to weeks) combined with our
smoothed resistivity models limited our ability to capture subtle diel temperature transience across discrete fractures
or flow features. Although the electrical resistivity method was not able to definitively resolve a groundwater-





surface water mixing zone, these data do provide insight into the magnitude, lateral extent and spatiotemporal scale
of geoelectrical transience within the upper few meters of rock.

**6.0   Conclusions**

Electrical resistivity methods were used to investigate the temporal geoelectrical response beneath a bedrock river
within the upper 6 meters of rock over a full annual cycle.  Induced resistivity measurements across the 200 m reach
of the river during high and low flow conditions showed that spatiotemporal variations were dependent upon
riverbed morphology.  Time-lapse electrical resistivity imaging of the pool and riffle portion of the river, sampled on
a semi-daily to semi-weekly interval, showed consistently higher resistivity at the pool with more elevated
resistivities along the south shoreline.  The formation of a transient 2 m thick low-resistivity zone within the pool in
mid-summer appears to be associated with an increase in surface water/bedrock formation temperatures during
seasonally low river flux.  Conversely, the riffle was characterized by a 3 m thick low-resistivity zone spanning the
entire width of the river, underlain by a more resistive material.  These lower resistivities at the riffle suggest the
presence of more porous bedrock material consistent with enhanced-dissolution of rock and/or a layer of weathered
bedrock zone overlying competent rock.
Although seasonal geoelectrical dynamics were observed at both the pool and riffle, the pool was more transient and
exhibited a broader range, yet spatially more uniform distribution in resistivity. Conversely, the riffle exhibited more
lateral variability in resistivity along across the riverbed.   Seasonal cooling was accompanied by a higher-resistivity
zone emanating from the south shore to north shore in both the pool and riffle.  This resistivity trend reversed during
the seasonal warming cycle, becoming less-resistive toward the south shoreline as seasonal temperatures increased
and river flow decreased.  The formation of ground frost and basal ice along the riverbed had a strong and
sometimes negative impact on the seasonal resistivity profiles during the winter months.  Intraseasonal geoelectrical
transience associated with major precipitation events, which were accompanied by short-period perturbations in
surface water temperature and specific conductance had a relatively small impact on riverbed resistivity.  This may
be explained by the presence of a strong groundwater discharge zone across this reach of the river, which may have
limited or moderated the electrical resistivity changes within the suspected groundwater-surface water mixing zone**.**
Time-lapse temperature profiling within the angled borehole underlying the river revealed active groundwater flow
zones. While our resistivity measurements captured geoelectrical dynamics within the upper few meters of riverbed,
these data are indirect evidence of a groundwater surface water mixing zone; whether the observed geoelectrical
transience are primarily a function of seasonal temperature fluctuations or transience in ionic concentration, in
response to precipitation events, will require further investigation. Nevertheless, our study demonstrates that surface
electrical resistivity has the capacity to detect and resolve changes in electrical resistivity within a bedrock riverbed.
Given the complex fracture distribution, geometry and dissolution features inherent to sedimentary rock, surface
resistivity methods may be most effective in the initial design and placement of more direct measurement methods,
thereby reducing instrumentation costs and impacts to ecosensitive environments.





**DATA AVAILABILITY**
The data used in this study are presented in the figures. Complete monitoring data sets (Figures 10 and 11) and can
be made available upon request from the corresponding author.

**TEAM LIST**
**Steelman, Kennedy, Capes, Parker**

**AUTHOR CONTRIBUTION**
**Steelman** designed the experiment, conducted the surveys, and analysed the geophysical data; **Kennedy** designed
the borehole network, logged the core, instrumented the hydrological monitoring network; **Capes** designed and
collected the temperature profiles, logged the core and supported hydrological data collection and interpretation;
**Parker** contributed to the design of the hydrological geophysical monitoring network, and supported conceptual
understanding of groundwater flow through fractured rock.

**COMPETING INTERESTS**
The authors declare that they have no conflict of interest.

**ACKNOWLEDGEMENTS**
This research was made possible through funding by the Natural Sciences and Engineering Research Council of
Canada in the form of a Banting Fellowship to Dr. Steelman, and an Industrial Research Chair (Grant #
IRCPJ363783-06) to Dr. Parker. The authors are very appreciative of the field contributions of staff and students at
the University of Guelph and University of Waterloo, particularly field-technicians Dan Elliot and Bob Ingleton.
This work would not have been possible without land-access agreements with Scouts Canada, Ontario Ministry of
Natural Resources, and river-flow data provided by the Grand River Conservation Authority.





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





777                                                          **Figures**





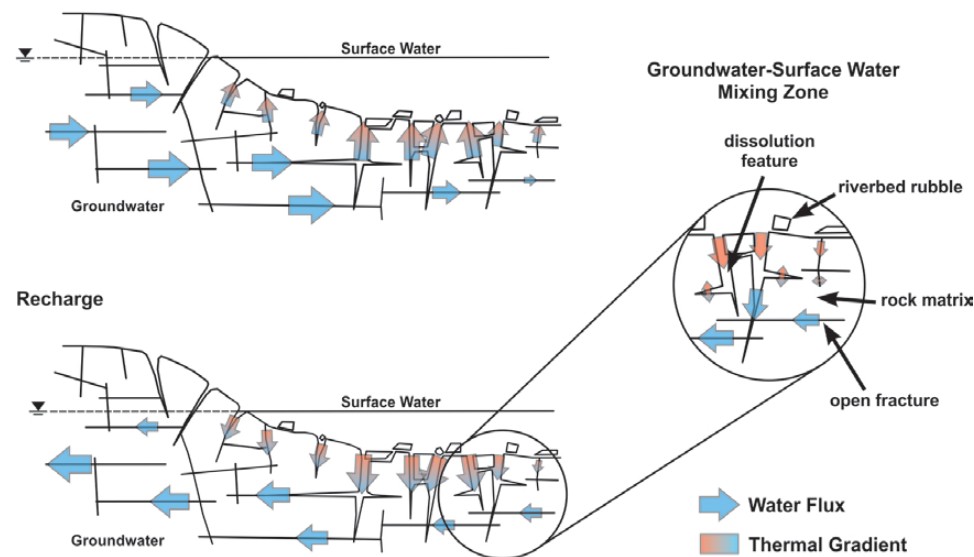


Figure 1: General conceptual model of the groundwater flow system beneath a fractured bedrock river.
Groundwater-surface water mixing is controlled by open fractures and dissolution-enhanced features with secondary
exchanges (flux or diffusion) occurring between fractures and rock matrix.





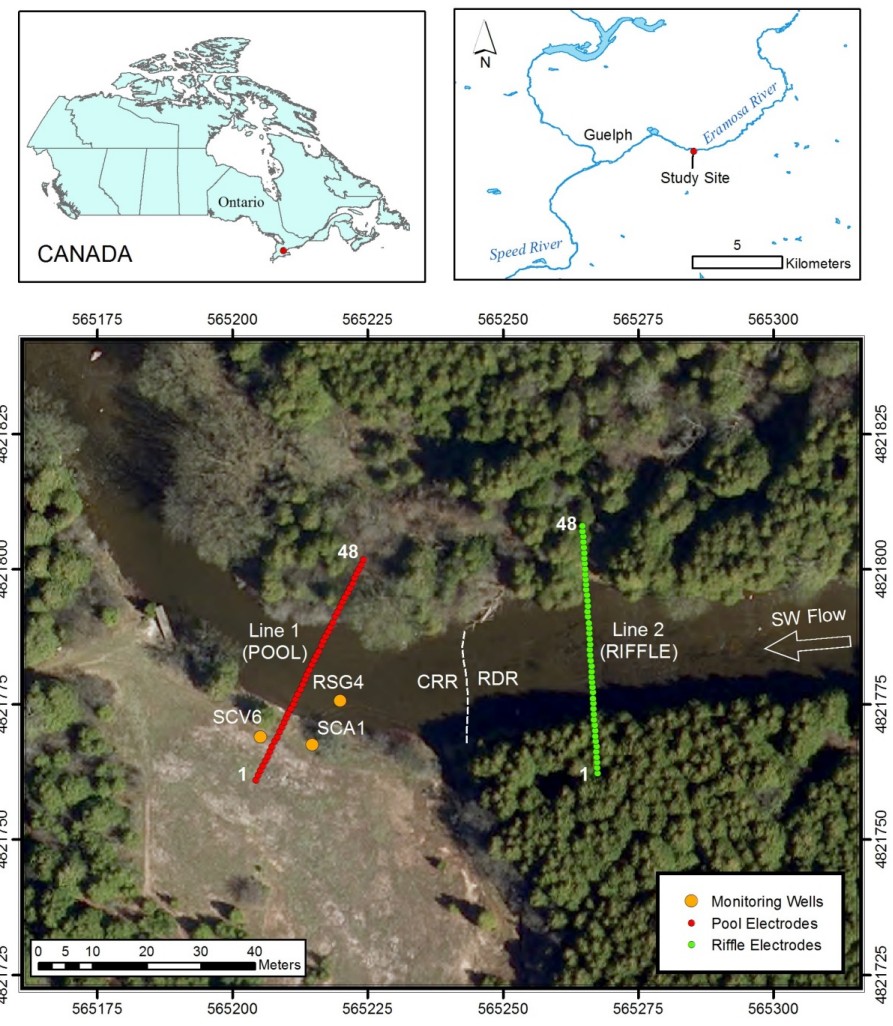


Figure 2: Field site located along the Eramosa River near Guelph, Ontario, Canada. Corehole and monitoring points
are shown with fixed electrical resistivity transects located over a pool and riffle. The riverbed is described as either
rubble dominated riverbed (RDR) or competent rock riverbed (CRR) surface.



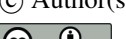






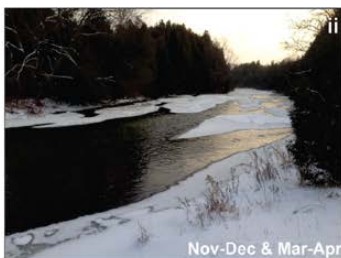
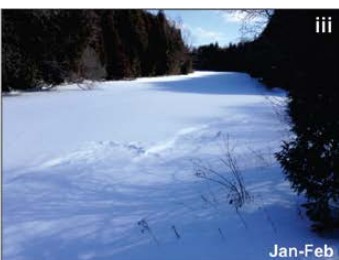

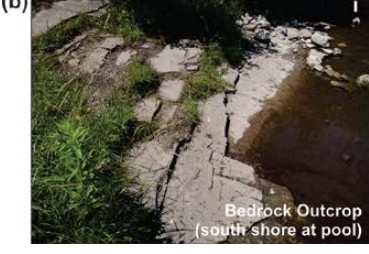
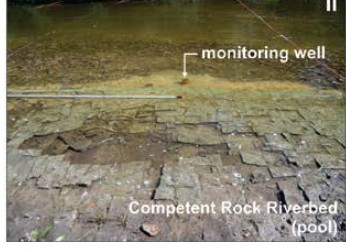
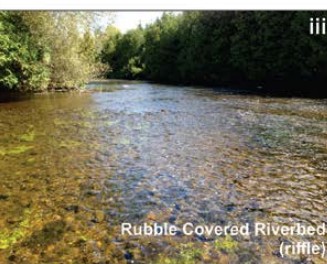


Figure 3: (a) Images of the river during monitored study period. (b) Examples of vertical and horizontal fracturing
within pool and rubble covered portions of the riverbed bedrock.











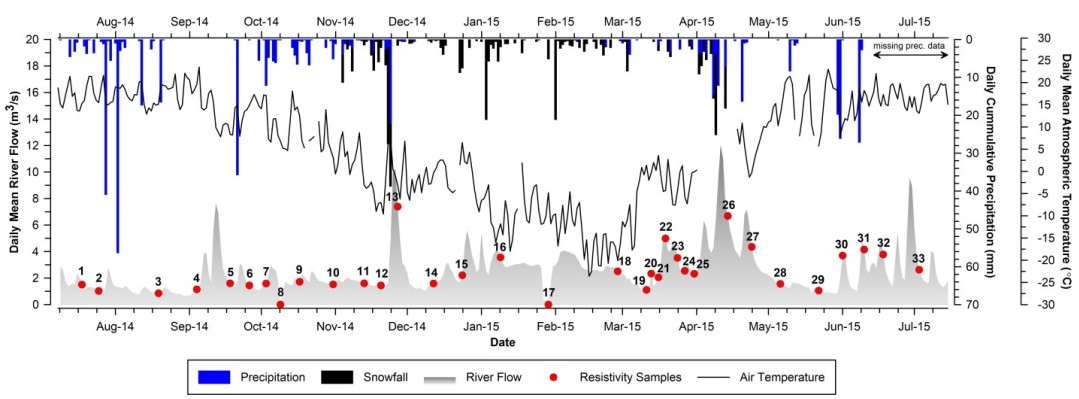


Figure 4: Continuously monitored atmospheric conditions and river flux from Watson Gauge during the study period
with discrete geophysical measurement events between 18-Jul-2014 and 3-Jul-2015.








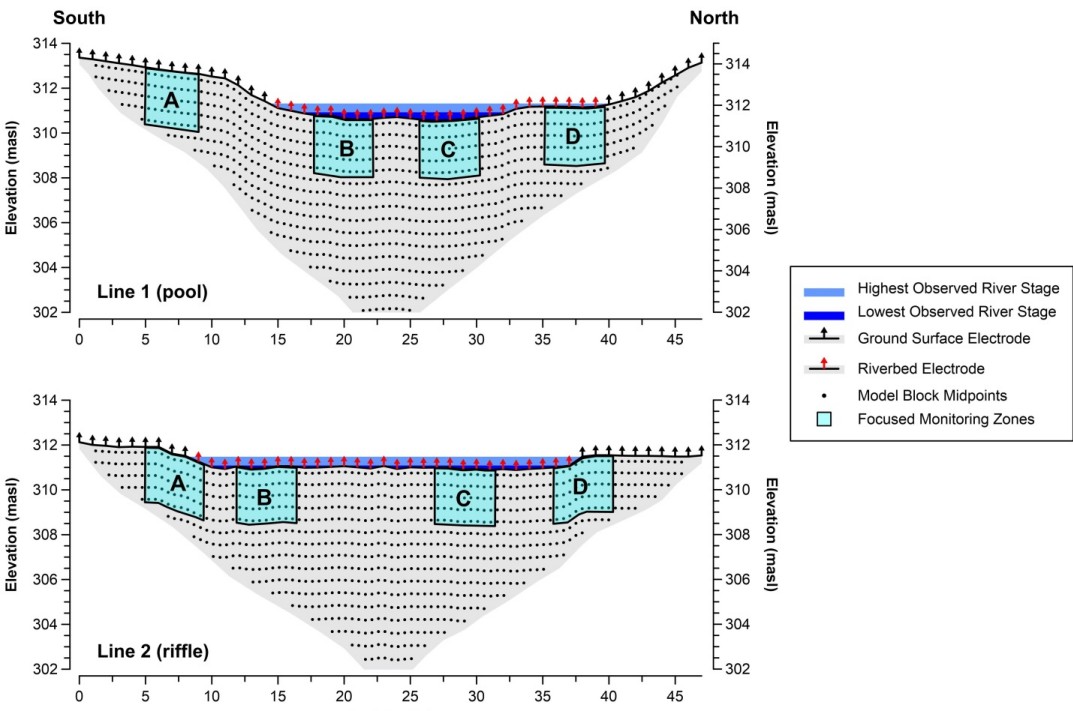


Figure 5: Electrical resistivity model set-up for the pool and riffle incorporating topographic variations, submerged
electrodes, river stage and surface water resistivity variations.  Model block points correspond to the interpreted
portion of the bedrock beneath the river. Zones A, B, C and D represent areas of focused monitoring beneath the
riverbed interface.








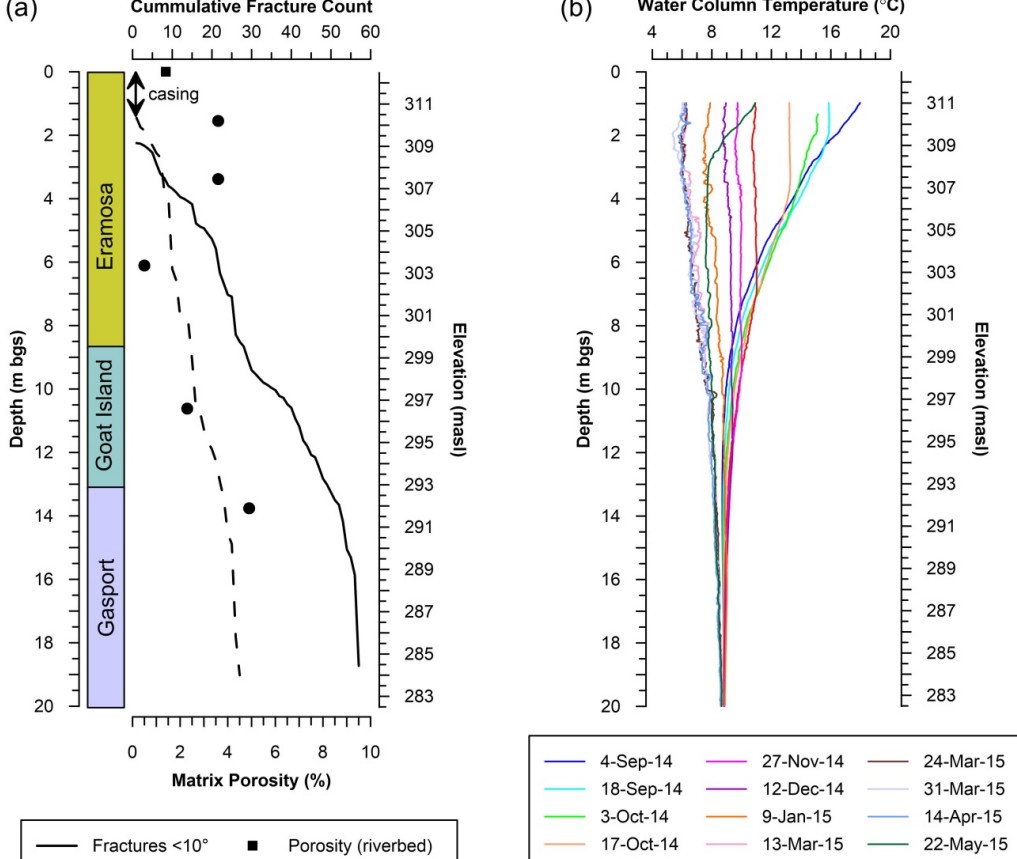


Figure 6: (a) Interpreted rock core from SCA1 (angled corehole plunging at 60° with an azimuth of 340°). Fracture

frequency and orientations were obtained using an acoustic televiewer log, while matrix porosity measurements

were obtained from subsamples of the continuous core. (b) Corehole temperature profiles of the SCA1 Flute™

sealed water column.






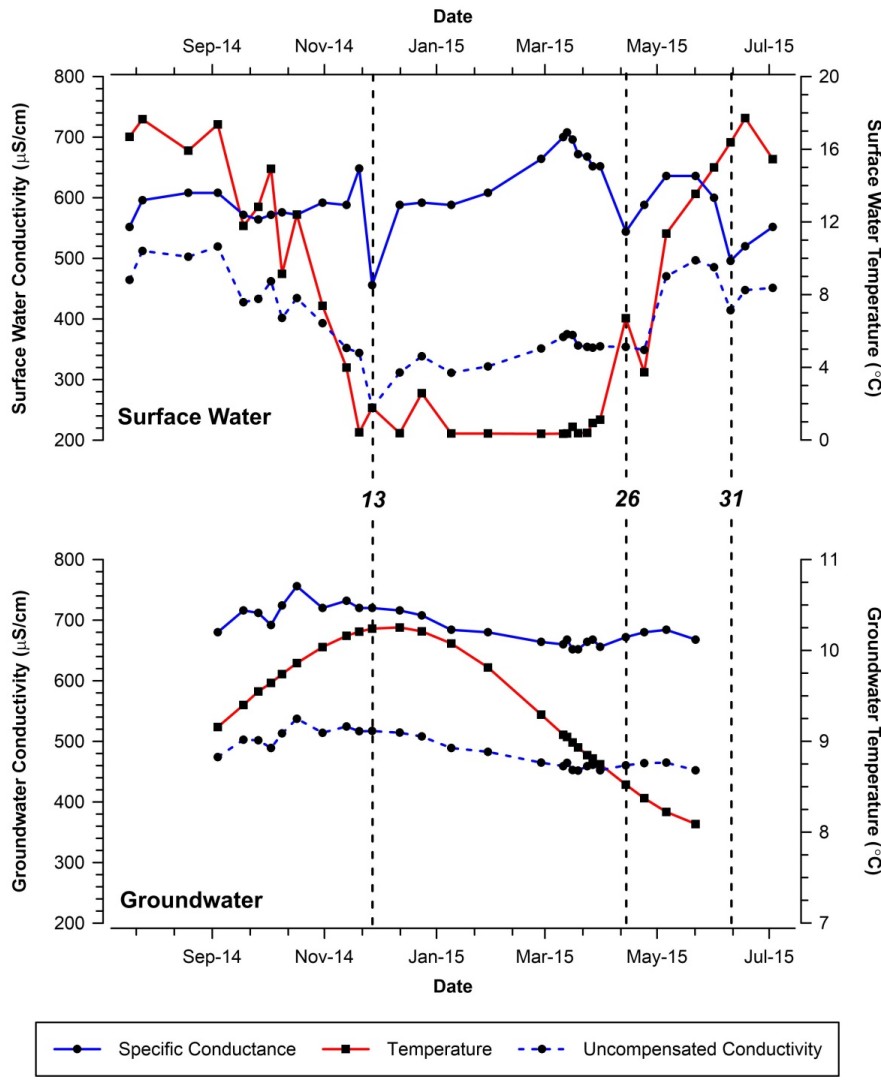


Figure 7: Specific conductance, temperature and uncompensated aqueous conductivity of surface water at RSG4 and
groundwater at the bottom of SCV6 (10.5 m bgs).  Uncompensated conductivity represents the actual conductivity
of the porewater after re-incorporating the effect of temperature using the sensors internal temperature-conductivity
correction factor.








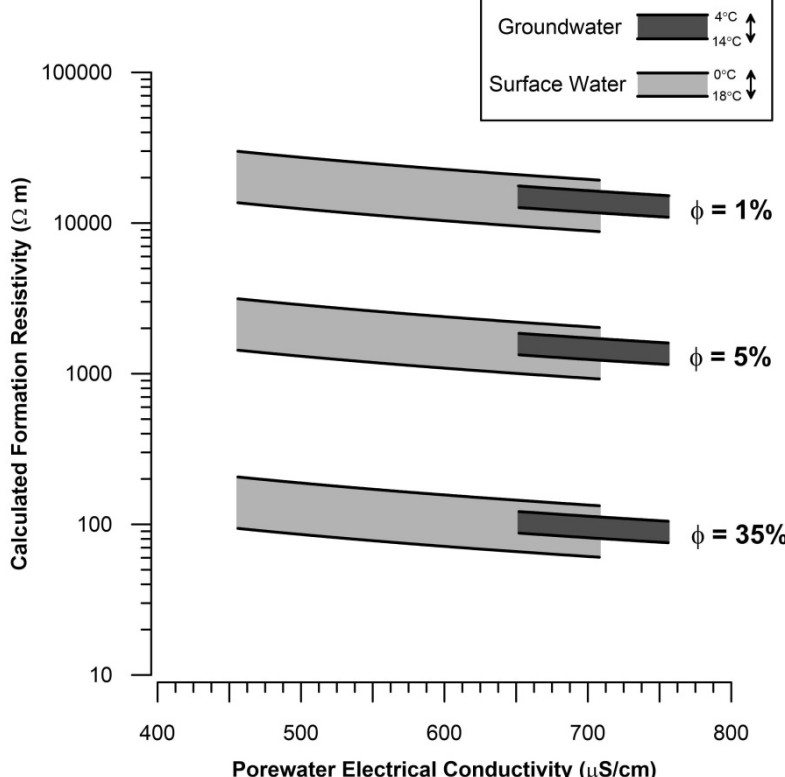


Figure 8: Calculated formation resistivity based on Eq. (1) and measured variations in surface water and
groundwater electrical conductivity including potential temperature effects based on Eq. (2).  A cementation factor
of 1.4 was used to represent the fractured dolostone bedrock.  Measured water conductivity and temperature were
obtained from CTD-Diver™ sensors deployed in RSG4 (surface water) and SCV6 (groundwater at a depth of 8 m
bgs), and the continuous RBR™ temperature profiles shown in Fig. 6b.  These data show the potential range in
formation resistivity based on the measured range in specific conductance and temperature for three different
porosity values.  Porosities of 1 % and 5 % correspond to the range measured in the core, while a value of 35 %
would be representative of a rubble zone.







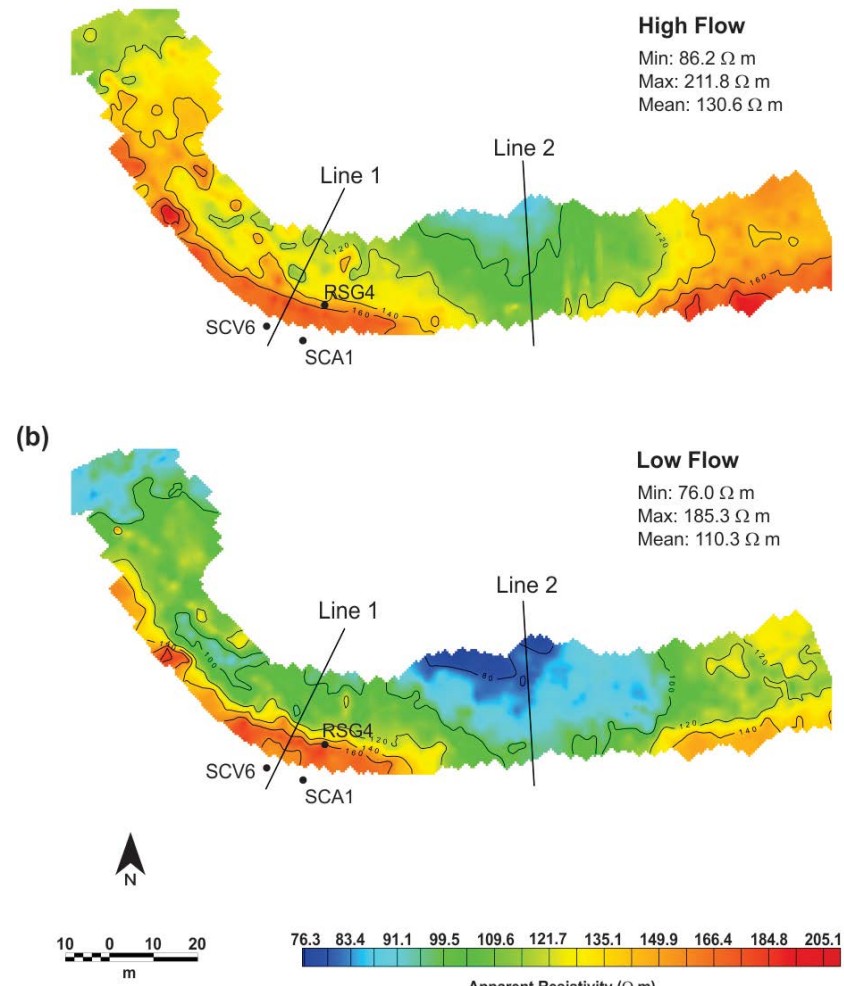


Figure 9: Riverbed resistivity obtained using an EM-31 ground conductivity meter during (a) high-flow/high-stage
conditions on 3-Apr-2013 and (b) low-flow/low-stage conditions on 7-Jul-2014.




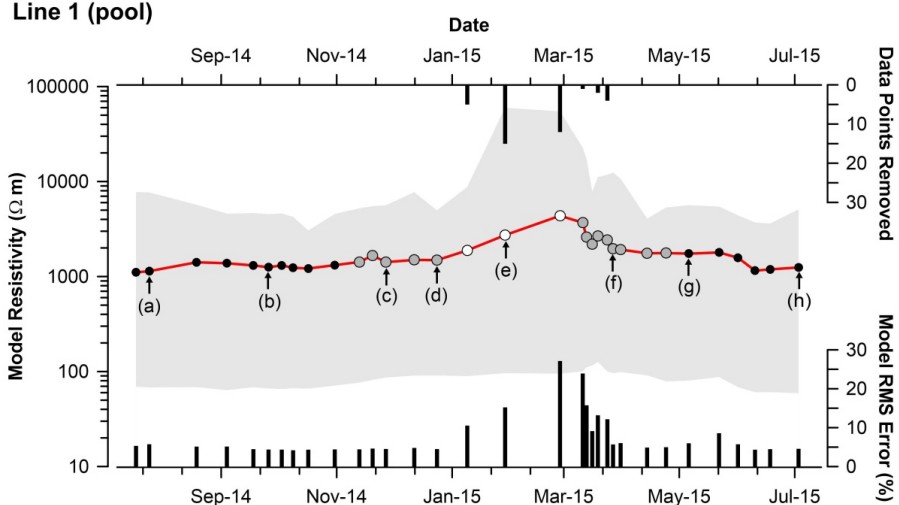

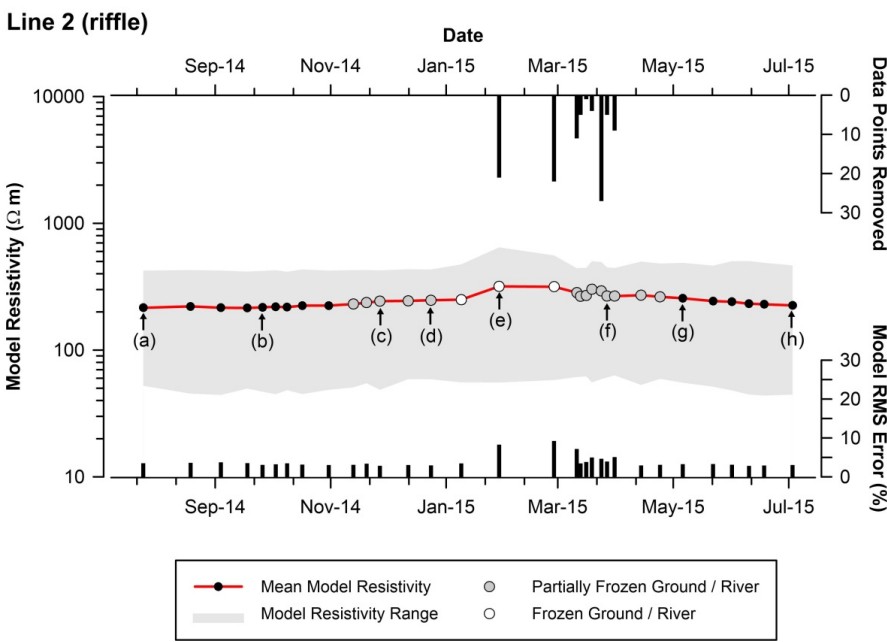


Figure 10: Temporal variations in inverted resistivity models for pool and riffle. Black dots represent unfrozen

conditions, grey dots indicate partially frozen conditions, while white dots indicate comply frozen river conditions.

Select resistivity models (a-h) along the time series are shown in Fig. 11.




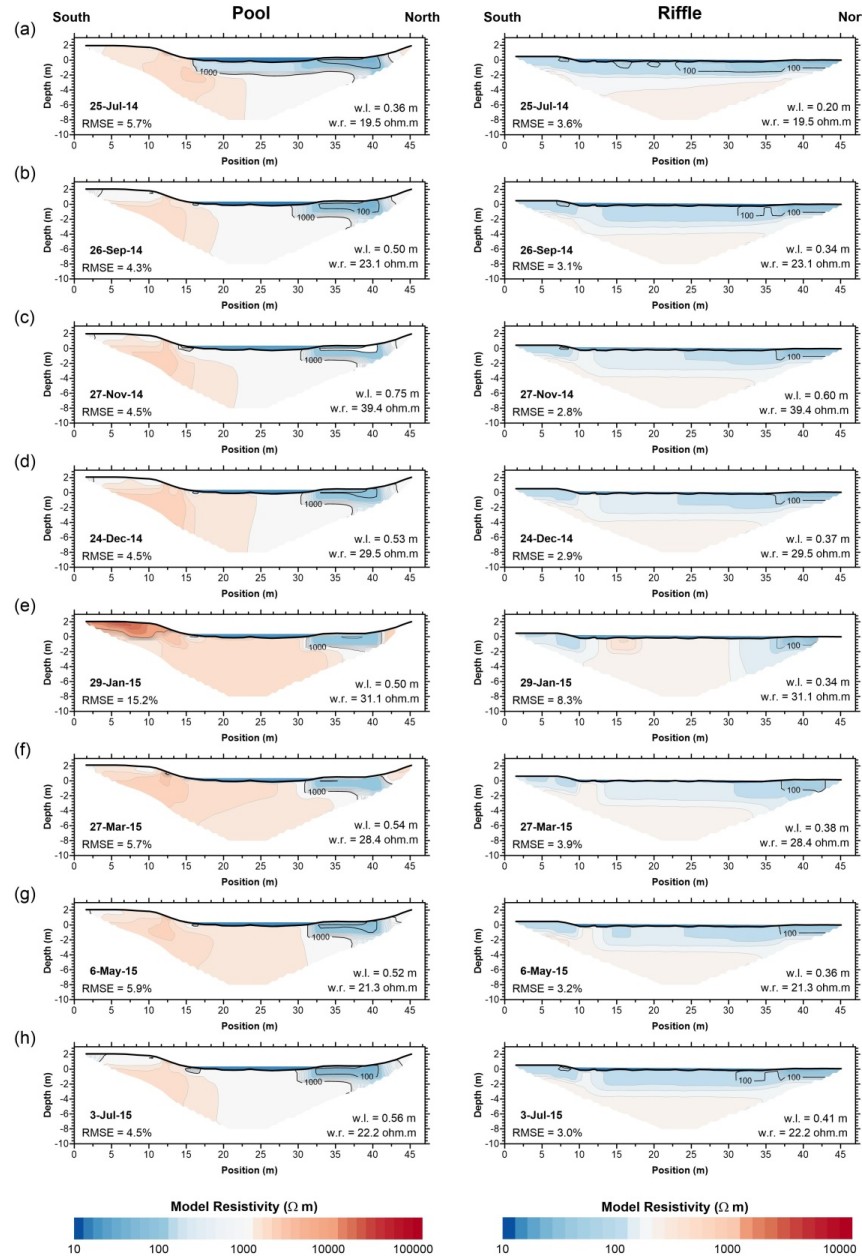


Figure 11: Representative inverse resistivity models across the pool and riffle orientated from south to north.
Datasets (a-h) are identified in Fig. 10.  River stage (w.l.) and surface water resistivity (w.r.) values were fixed in the
inverse model.  A marked increase in resistivity was observed beneath the river during colder seasonal conditions
(November through March), while lower resistivities were observed during warmer seasonal conditions (July).



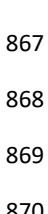





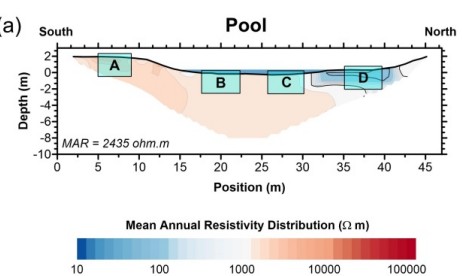
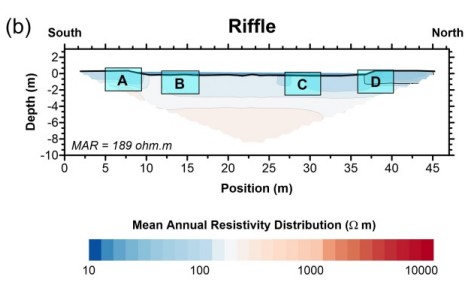

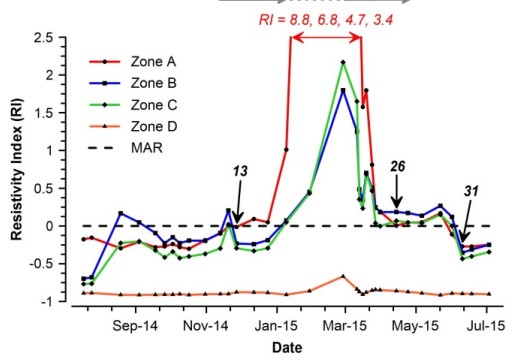
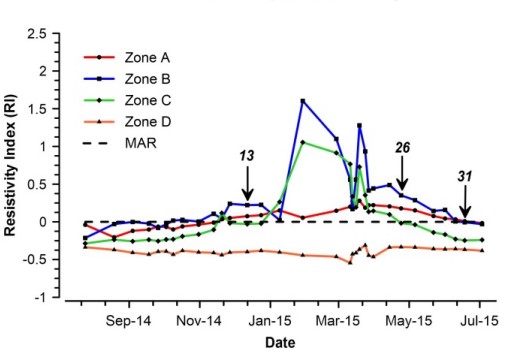


Figure 12: Spatiotemporal fluctuations in resistivity within the focused monitoring zones A, B, C and D. The
resistivity index (RI) was calculated using Eq. (3), using the mean zone resistivity (MZR) for a given measurement
date and the mean annual resistivity (MAR) of the whole profile.