# Peer review of "Electrical Resistivity Dynamics beneath a Fractured Sedimentary Bedrock Riverbed in Response to Temperature and Groundwater/Surface Water Exchange"

_Hydrology and Earth System Sciences, 2016_

## Referee Comment (RC1) · Anonymous Referee #1 · 3 Dec 2016

Review Electrical Resistivity Dynamics beneath a Fractured Sedimentary Bedrock Riverbed in Response to Temperature and Groundwater/Surface Water Exchange

The study deals with fractured sedimentary bedrock riverbeds and its spatio-temporal groundwater surface water exchange of the Eramosa River within the Grand River Watershed, Ontario, Canada. Surface electrical methods (ERT and EMI) were used for a quasi non-invasive assessment of the scale and temporal variability of riverbed temperature and groundwater-surface water exchange beneath its sedimentary bedrock riverbed. Underpinned were the solid geophysical data sets by a network of boreholes

and streamed piezometers, installed across the site. The study further contained highly relevant material for HESS, is well written, structured and citied and attested the authors' fundamental background and experiences with the selected topic. Fractured sedimentary bedrock systems and its interaction processes are very complex and difficult to descript. Hence the presented study provided a useful approach for cost-efficient investigation into the river flow regime. Still there are a few issues that needs to be addressed until the manuscript is ready for publication. Although the study impressed be the amount of high quality data and different applied methods, it lacks a little of a clear and comprehensive purpose. Studies of interaction flow processes at rivers are relatively common since about one decade, however the presented geophysical investigation at bedrock rivers combined with ground truths core data is unique. Moreover, the economic and ecological potential of global bedrock rivers are remarkable, hence the study can be considered as a pioneer study with portable conclusions. This seems the strengths of this study, however it needs to be emphasized. It would be helpful if the authors establish somehow a relation between their investigations and findings, and a direct affected related system, such as a water supply or ecosystem services. If not available for the selected test site then relevant literature can be used. In this respect the manuscript might be slightly rearranged. Introduction should contain a clear state of the art containing the situation, the problem, the challenge and the provided response (solution). Although I like the extra paragraph 2 'background', it's quite uncommon and could be incorporated into the introduction and the material and method part, respectably. It is always helpful to read what other studies have archived at similar test sites and / or with similar approaches and where they were limited. This should be highlighted in the introduction and in the conclusion (and in the abstract too). The conclusion should hence contain the extracted information given by the findings as kind of a 'take home message' for the scientific community rather than a second abstract. I recommend acceptance with minor revision. Below a few remarks: - Line 94 – 96 amount of references can be decreased by a related review - Line 126 please mention that the presented Archie's law is simplified are provide the whole equation - Line 144

just mentioned the temperature correction was done by Arps (1953) might be sufficient - Line 224 indicate the approx. location of the EMI lines in Figure 2 - Line 270 use 'Fig' or Figure either but consisted - Line 291 matrix porosities from the corehole relatively low in respect to? A short reference value for the same rock material from the literature is useful or do you mean in comparison to the weathered or broken rubble zone? If so, please mention it - Line 321 see above - Line 328 it is hard to follow here or maybe I missed the point how you ended up with the 46%, according to Archie in Eq.1? How do you get the Sigma w values or am I totally off? If you re-arrange the equation it needs to be mentioned if not showed - Line 332 – 349 how could you be sure that the EMI data were not affected by outside conditions, do you temperature corrected the ECa as well? - Line 420 I prefer including of the discussion together with the presentation of the results. This helps to shorten the manuscript which is almost every time helpful - Line 857 since you presented the ERT results in Figure 11 by common scale, Figure 10 is kind redundant, mention in the text that the ERT data quality (RMSE, removed data points) were higher under frozen, partly frozen conditions

———————————————

---

## Referee Comment (RC2) · Anonymous Referee #2 · 25 Jan 2017

This manuscript is on the topic of geophysical and traditional measurements of a reach of river to investigate the suitability for time lapse ERT to study river bottom processes. The writing is in clear, good English, and the figures are mostly very readable and nicely drafted. The topic – either from the Hydrology or Geophysics perspective – certainly has the potential to be of interest to HESS readership. I believe the topic of this work fits into the scope of this journal. The most significant limitation I see to this work is related to the experimental design, which is largely absent from the writing. In short, it is difficult to tell what was being tested about the hydrology, and why measurements were implemented to carry out that test. The stated hypothesis is apparently related to "will

the geophysics work," while the theme of riverbed processes appears and disappears throughout the manuscript. In the end, I remained confused about exactly what the reader was meant to take away from this given the setup of the writing and the design of the study. There is certainly lots of good data here and on some level this has the potential to be of high interest to the hydrology community, but there is a need for substantial revision for focus. There were several other notable issues/limitations related to measurement methods, data processing, and absence of some measurements that are detailed in my General and Line-by-line comments below. At this time I am recommending this manuscript be returned for major revision, however if the experimental design is not substantially clarified and the focus reworked to highlight hydrological interpretations, a second review would likely not result in a favorable recommendation.

General Comments:

There is a substantial disconnect between the topic of the science question and the posed hypotheses. Although the science question is not explicitly stated, it is my best interpretation that the following reflects the intent of inquiry: "...there remain gaps in our conceptual understanding of groundwater-surface water interaction and exchange mechanisms in bedrock rivers where discrete fracture networks will dominate groundwater-surface water flux with secondary interactions supported by the porous rock matrix." On the other hand, the hypothesis is explicitly stated, although it appears to be limited to a yes-or-no "will it work" type of speculation: "we hypothesize that a groundwater-surface water mixing zone – encompassing fracture and matrix flow and diffusion – may be identified within a fractured bedrock riverbed by monitoring spatiotemporal changes groundwater temperature and porewater electrical conductivity using minimally invasive electrical resistivity methods." Further complicating matters is the text between Line 69 and 76 that highlight the hydrological outcomes while disregarding the stated hypothesis.

Throughout the manuscript, speculative statements about river ice, river-bottom ice and frost are made, though they do not appear to be supported by any direct measure-

ments or observations.

Estimates of loss along reaches based on calculations of discharge using rating curves in conjunction with stage height monitoring appear to be absent. This line of evidence would substantially help to support geophysical observations.

I felt that the following questions posed early on in the manuscript were not clearly answered: Do you find that groundwater-surface water interaction was restricted by poor vertical connectivity and limited
bedrock incision? Did you find that groundwater-surface water connectivity through discrete fractures was highly variable in space and time, and depended on fracture size or aperture, river stage, and the
distribution of hydraulic head within the flow system?

There is a huge amount of data contained in this manuscript, however in some cases data was left unused in interpretations and discussion. For example, precipitation & snowfall, daily river stage, fracture content as a function of depth, atmospheric temperature, etc. Why include these data if they are not utilized?

In the end, if the hypothesis was to test "will ERT work for this" I think that was not clearly answered, and furthermore, given the high dependence on temperature, it may be that that answer is "no."

Line-by-line Comments:

Line 55: Perhaps add a comment on what Fan et al., 2007 found here?

Line 63: There is a lot going on in the figure and it is weakly linked to the text. Are you testing these concepts?

Line 92: The Singha paper has 2014 printed on top of it, but I'm not sure which date is correct.

Line 175 – 180: Was the formation of basal ice actually observed at the site or only inferred?

Line 213 – 216: I am unfamiliar with this method of sampling temperature while the sensor is in motion. Certainly the sensor itself, however small it might be, has some thermal mass that requires time to equilibrate to the surrounding water temperature. Even though the sensor is capable of measuring at 0.5 s rate, that does not mean that the measured data are reflecting changes in the formation at that rate. A reliable reference should be included here to justify the method, and a controlled validation test and sensor calibration under laboratory conditions should be conducted to quantify sensor response.

Line 219: What is the value of measuring snowfall accumulation if snow density is not also reported with a conversion to SWE?

Line 221/Figure 4: What scale are the red dot "Resistivity Samples" on? They appear to be only temporal and unitless, however they seem to track the river stage which is confusing. Is 'snowfall" in this figure converted to SWE? If not, please do and clarify the label.

Line 228: "effective sensing depth" does this mean measurements are reflective of the 6m depth zone, or the entire aggregated zone 0 to 6m depth?

Line 237: "BLANKED by bedrock rubble" I am not familiar with this usage of "blanked" in this context. Suggest rewording for clarity.

Line 239 – 244: Does the electrode construction method have any particular importance to this study? This sounds like very typical ERT cable construction, albeit by the end-user rather then a professional fabricator. Probably could be deleted.

Line 261: How was the measurement time determined? It is known that diurnal fluctuations in stream water temperature may be of magnitude in excess of 10C, similar to your annual range of groundwater temperatures. Also you acknowledge the affect of temperature on the ERT readings; how does the timing of the measurements affect the data due to daily fluctuations?

Constantz, Jim, Carole L. Thomas, and Gary Zellweger. "Influence of diurnal variations in stream temperature on streamflow loss and groundwater recharge." Water resources research 30.12 (1994): 3253-3264.

Constantz, Jim. "Interaction between stream temperature, streamflow, and groundwater exchanges in alpine streams." Water resources research 34.7 (1998): 1609-1615.

Line 262: "manually filtered" What criteria was used for manually filtering? Why was this approach used rather than the common quantitative method of envelope filtering based on an error model?

Slater, Lee, et al. "Cross-hole electrical imaging of a controlled saline tracer injection." Journal of applied geophysics 44.2 (2000): 85-102.

Line 264: "moderate to high damping" – Does this mean different damping factors were used on each dataset? What is the numerical value of damping used and how is this value incorporated into your inversion scheme?

Line 267: What parameters, how were they optimized, and were identical settings used for all datasets?

Line 268 – 269: Certainly achieving the lowest possible RMS is not the optimal approach to achieve the most "believable" geophysical result. How does the RMS relate to observed measurement noise/errors? At what point is the inversion fitting noise?

Line 280: Is there a reference for this Resistivity Index? What is the justification for manipulating the data in this way?

Line 293 – 294: Where is the data demonstrating upward head shown?

Line 307/Figure 6b: It would be helpful to grade the colors of the lines linearly to more easily show the temperature trend. Even better would be to present these data as a matrix/grid where time is on the x-axis, depth is on the y-axis, and color represents temperature.

Line 308: "correspond to areas" I cannot tell from the figure how the fracture patterns correspond to the temperature results. Perhaps some annotation, or another approach to presenting these data would help.

Line 321 – 331: [Figure 8] This seems more like a discussion point rather than a result.

Line 356 – 358: "greater number of measurements..." why would the number of removed data cause higher RMS? Presumably if the data were removed, they would no longer be included in the RMS calculation.

Line 374: What are the observed thicknesses of basal ice and floating ice?

Line 415-416: "groundwater discharge in this section" I don't follow the logic why the relationship between substrate resistivity and "surface water response" indicates magnitude of discharge that could be interpreted in this way. Also, correlation is not demonstrated or quantified.

Line 432: "strong upward hydraulic gradients" please indicate where this is demonstrated by data.

Line 436: "likely dominated the bulk electrical response" Why 'likely'? Based on the evidence shown, temp is clearly dominating the ERT signal.

Line 445: Where is ground frost or riverbed ice formation measured data shown?

Line 459 – 473: As previously stated, data showing the frost and ice should be shown.

474: I'm not sure what evidence directly supports this statement. The provided sensitivity analysis appears to only vary the river water electrical properties; this doesn't seem to directly simulate the presence of ice as claimed in this statement.

479 – 480: Quantify this? Why would inputting a one-half of true river water resistivity lead to "substantial overestimates of river resistivity" – wouldn't the river water resistivity be fixed so that the output = the input?

[Figure]

486: What about a synthetic model example?

510: How does geoelectrical transience translate into hydrological processes?

Line 511 – 452: The conclusions section contains substantial summary and could be reworked for improved focus.

Figure 5: The purpose of this figure is unclear and I suggest that it could be deleted. The A/B/C/D locations are already indicated on Figure 12; the river stage information is presented on Figure 4; the location of the model block midpoints does not appear to be substantially important to the manuscript.

Figure 9: Perhaps showing only the difference between these two maps would make interpretation easier? If not only difference, then perhaps just adding a third difference panel. Also, isocontour labels are too small to read.

Figure 10: What is the model error relative to the measurement errors? What is the purpose of showing these vast bulk averages when that eliminates any of the valuable spatial information yielded by using tomographic methods? Figure 12 seems to be much more useful than this.

Figure 11: Very nice layout and presentation of this figure, however certainly this needs to be replotted to show the difference between (b) through (h) relative to (a) in both columns.

---

## Author Comment (AC1) · 7 Feb 2017

**Author Response to Reviewer #1:**
**Hydrol. Earth Syst. Sci. Discuss., doi:10.5194/hess-2016-559, 2016.**

First and foremost, we would like to thank the reviewer for donating their time to review our manuscript and for providing a fair and constructive review of our work. These comments and suggestions will undoubtedly improve the impact and utility of our paper.

**Reviewer 1: General Comments**

The study deals with fractured sedimentary bedrock riverbeds and its spatio-temporal groundwater surface water exchange of the Eramosa River within the Grand River Watershed, Ontario, Canada. Surface electrical methods (ERT and EMI) were used for a quasi non-invasive assessment of the scale and temporal variability of riverbed temperature and groundwater surface water exchange beneath its sedimentary bedrock riverbed. Underpinned were the solid geophysical data sets by a network of boreholes and streambed piezometers, installed across the site. The study further contained highly relevant material for HESS, is well written, structured and citied and attested the authors' fundamental background and experiences with the selected topic. Fractured sedimentary bedrock systems and its interaction processes are very complex and difficult to describe. Hence the presented study provided a useful approach for cost-efficient investigation into the river flow regime. Still there are a few issues that needs to be addressed until the manuscript is ready for publication. Although the study impressed be the amount of high quality data and different applied methods, it lacks a little of a clear and comprehensive purpose.

Response: We appreciate the reviewer's encouraging assessment of our work, and agree that there are areas that will require clarification and improvement prior to publication. We acknowledge that the motivation for this work was not clearly defined in the introduction. We can definitely improve the introduction by clearly stating the problem, followed by objectives and main contributions of this work.

Studies of interaction flow processes at rivers are relatively common since about one decade; however, the presented geophysical investigation of bedrock rivers combined with ground truths core data is unique. Moreover, the economic and ecological potential of global bedrock rivers are remarkable, hence the study can be considered as a pioneer study with portable conclusions. This seems the strengths of this study, however it needs to be emphasized.

Response: We believe the introduction can be strengthened by highlighting these points raised by the reviewer.

It would be helpful if the authors establish somehow a relation between their investigations and findings, and a direct affected related system, such as a water supply or ecosystem services. If not available for the selected test site then relevant literature can be used. In this respect the manuscript might be slightly rearranged. Introduction should contain a clear state of the art containing the situation, the problem, the challenge and the provided response (solution).

Response: We can improve the problem statement through more explicit discussion of regional implications of groundwater surface water interaction within the context of bedrock aquifers. Although a direct comparison with the regional hydrogeologic system will be beyond the scope of this paper, we can definitely improve the context of this work through a more thoughtful assessment of the local groundwater problems. We can also make substantial improvements to the last three paragraphs of the introduction by clearly stating the problem, the challenges/conceptual limitations, and what our paper contributes.

Although I like the extra paragraph 2 'background', it's quite uncommon and could be incorporated into the introduction and the material and method part, respectably. It is always helpful to read what other studies have archived at similar test sites and / or with similar approaches and where they were limited. This should be highlighted in the introduction and in the conclusion (and in the abstract too).

Response: We agree that some of the geophysical background information on rivers could be integrated into the introduction. Similarly some of the electrical properties discussion should be placed in the methods section. At this point we can definitely agree to consider some reorganization of text as suggested by the reviewer.

The conclusion should hence contain the extracted information given by the findings as kind of a 'take home message' for the scientific community rather than a second abstract. I recommend acceptance with minor revision.

Response: We agree with the reviewer. We believe a reorganization and enhancement of the introductory text (i.e., current state of the science, more definitive problem statements, and the specific contributions to our work to the advancement of the conceptual model) will set the stage for a more appropriately written conclusion section.

**Specific Comments:**

Line 94 – 96: amount of references can be decreased by a related review

Response: There are only reference 10 studies, which isn't exhaustive considering the broad topic.

Line 126: please mention that the presented Archie's law is simplified are provide the whole equation

Response: We will make the necessary revisions.

Line 144: just mentioned the temperature correction was done by Arps (1953) might be sufficient

Response: Since the equation was used with our results (Figure 8) we think it might be best to keep the equation. We will consider moving this "background" material to the methods section.

Line 224: indicate the approx. location of the EMI lines in Figure 2

Response: We generally agree with the reviewer but the result is visually distracting in a paper. We do state in the text the orientation and spacing of the EMI survey lines and we think this is sufficient.

Line 270: use 'Fig' or Figure either but consisted

Response: We believe the full form of the word is used at the start of a sentence. We will ensure the final document is properly formatted.

Line 291: matrix porosities from the corehole relatively low in respect to? A short reference value for the same rock material from the literature is useful or do you mean in comparison to the weathered or broken rubble zone? If so, please mention it

Response: We can remove our qualitative referencing to avoid potential confusion/distraction. Porosities in these sedimentary particular rocks can range from <1 to >15% (i.e., matrix porosities). Inclusion of fractures and dissolution features can result in even greater and more variable porosities. We believe our description is accurate and appropriate in this section.

Line 321: see above

Response: see above.

Line 328: it is hard to follow here or maybe I missed the point how you ended up with the 46%, according to Archie in Eq.1? How do you get the Sigma w values or am I totally off? If you re-arrange the equation it needs to be mentioned if not showed

Response: The percent fluctuation represents the potential influence of temperature based on the observed ranges in groundwater and surface water. The temperature data is shown in Figure 7. Aqueous resistivity in Eqn. 1 is calculated using Arps law (Eqn 2). So the plot in Figure 8 simply shows the potential range in formation resistivity for the observed groundwater temperatures within the river bed. The % change stated in the text is the variability that is plotted (i.e., between isotherms 4-14 °C and 0-18°C) in Figure 8. We can adjust the phrasing in this section to improve clarity.

Line 332 – 349: how could you be sure that the EMI data were not affected by outside conditions, do you temperature corrected the ECa as well?

Response: Assuming the reviewer is referring to atmospheric temperature effects on the instruments performance, we do not believe, nor does the manufacturer report that ECa measurement accuracy of the EM-31 (+/- 5% at 20 mS/m) would have any meaningful effect on the reading.  The ECa measurements we report in Figure 9 are, of course, sensitive to the temperature of the media as indicated by Eqn 2 and illustrated in Figure 8. ECa variations in the rock are expected to be associated with either temperature fluctuation or specific conductance of the fluid.  Temperature fluctuations within the rock will indicate a zone of influence or transience. This could be due to differences in seasonal groundwater-surface temperature or mixing or changes in groundwater chemistry. The data we present identifies measurable changes in the geoelectrical conditions beneath the riverbed, whether it's temperature or EC based or some combination of the two; regardless, the spatial extent of these dynamics support the existence of a mixing zone or simply a zone of thermal variability in the rock unique to the river flow system.

Line 420: I prefer including of the discussion together with the presentation of the results. This helps to shorten the manuscript which is almost every time helpful

Response: We appreciate the reviewer's preference here. But considering the nature of the discussion in this paper we felt it best to separate it from the results to avoid potential confusion (i.e., geophysical results vs. hydrogeophysical interpretation).

Line 857: since you presented the ERT results in Figure 11 by common scale, Figure 10 is kind redundant, mention in the text that the ERT data quality (RMSE, removed data points) were higher under frozen, partly frozen conditions

Response: While we understand the reviewer's comment, we believe the information presented in Fig. 10 is highly informative. The study does contain an immense amount of information, which cannot be presented or summarized in its entirety, but the seasonal trends need to be presented in some way. The unique field conditions resulted in many challenges in data processing and interpretation. To our

knowledge, there has been no similar study performed in a river in this type of environment, let alone a bedrock rock river.

One of the unique contributions of our study is the results associated with the seasonal freeze-up and thaw of the river. This period presented many challenges, with respect to data collection, data analysis and interpretation. Yet, to ensure that this was not overlooked we wanted to provide a reasonable overview of the data trends and the impacts of varying site conditions on the measurement and analysis of data. Therefore, Figure 10 provides a critical overview of the modeled data partially shown in Figure 11 and Figure 12. This includes the data points removed from the models due to site conditions, resulting in the variable RMS errors. The Figure also provides an overview of the trends in each model (median, min, max) over the annual cycle at each location along the river. This figure also serves as a temporal reference for the models presented in Figure 11, which we believe is very important.  Figure 10 also shows the seasonal limitations of the ERT method, which we believe is a significant contribution to the methodology. Therefore, we remain supportive the information summarized in Figure 10.

---

## Author Comment (AC2) · 7 Feb 2017

**Author Response to Reviewer #2:**
**Hydrol. Earth Syst. Sci. Discuss., doi:10.5194/hess-2016-559, 2016.**

First and foremost, we would like to thank the reviewer for donating their time to review our manuscript and for providing a fair and constructive review of our work. These comments and suggestions will undoubtedly improve the impact and utility of our paper.

Reviewer 2: General Comments

This manuscript is on the topic of geophysical and traditional measurements of a reach of river to investigate the suitability for time lapse ERT to study river bottom processes. The writing is in clear, good English, and the figures are mostly very readable and nicely drafted. The topic – either from the Hydrology or Geophysics perspective – certainly has the potential to be of interest to HESS readership. I believe the topic of this work fits into the scope of this journal. The most significant limitation I see to this work is related to the experimental design, which is largely absent from the writing. In short, it is difficult to tell what was being tested about the hydrology, and why measurements were implemented to carry out that test. The stated hypothesis is apparently related to "will the geophysics work," while the theme of riverbed processes appears and disappears throughout the manuscript. In the end, I remained confused about exactly what the reader was meant to take away from this given the setup of the writing and the design of the study. There is certainly lots of good data here and on some level this has the potential to be of high interest to the hydrology community, but there is a need for substantial revision for focus.

Response: We believe the reviewer has provided a fair assessment. The problem statements and hypotheses we set out to test could be described more effectively in the introduction; Reviewer 1 raised a similar comment.  Because there were no previous examples of ERT being used to investigate riverbed dynamics in fractured rock, this study was in part, a type of proof of concept. We wanted to test the utility of ERT in a bedrock river environment under natural field conditions.  However, the motivation for this work was directly associated with furthering our understanding of potential mechanisms associated with a groundwater-surface water interaction in a fractured riverbed system. That being said, we believe we can significantly improve the introduction and associated text throughout the manuscript to ensure a clearer "takeaway message".

There were several other notable issues/limitations related to measurement methods, data processing, and absence of some measurements that are detailed in my General and Line-by-line comments below. At this time I am recommending this manuscript be returned for major revision, however if the experimental design is not substantially clarified and the focus reworked to highlight hydrological interpretations, a second review would likely not result in a favorable recommendation.

There is a substantial disconnect between the topic of the science question and the posed hypotheses. Although the science question is not explicitly stated, it is my best interpretation that the following reflects the intent of inquiry: ". . . there remain gaps in our conceptual understanding of groundwater-surface water interaction and exchange mechanisms in bedrock rivers where discrete fracture networks will dominate groundwater-surface water flux with secondary interactions supported by the porous rock matrix." On the other hand, the hypothesis is explicitly stated, although it appears to be limited to a yes-or-no "will it work" type of speculation: "we hypothesize that a groundwater-surface water mixing zone – encompassing fracture and matrix flow and diffusion – may be identified within a fractured bedrock riverbed by monitoring spatiotemporal changes groundwater temperature and porewater electrical conductivity using minimally invasive electrical resistivity methods." Further complicating matters is the

text between Line 69 and 76 that highlight the hydrological outcomes while disregarding the stated hypothesis.

Response: We appreciate the reviewers concerns and believe they can be addressed through reorganization and emphasis of key contributions.

Throughout the manuscript, speculative statements about river ice, river-bottom ice and frost are made, though they do not appear to be supported by any direct measurements or observations. Estimates of loss along reaches based on calculations of discharge using rating curves in conjunction with stage height monitoring appear to be absent. This line of evidence would substantially help to support geophysical observations.

Response: As stated in the manuscript the winter freeze-up period was accompanied by many challenges. In hindsight, a very different approach might have been used to fully understand the impacts of ice on the measurements, but we had not anticipated the conditions that we experienced. There was little that we could do to directly assess the ice in the river given the resources we had at our disposal. This is acknowledged in the paper. For this reason, the river ice is examined in the discussion rather than the results to avoid confusion between what was measured and what is interpreted. This paper focuses on the geoelectrical dynamics within the riverbed; we do not attempt to provide analysis of larger-scale hydrologic flow system. Estimates of loss/gain were measured at this site but are being prepared in a separate hydrological paper by other workers. Future work may integrate these data sets but at this time we remain focused on the geophysical transience and processes occurring with the upper few meter of rock.

I felt that the following questions posed early on in the manuscript were not clearly answered: Do you find that groundwater-surface water interaction was restricted by poor vertical connectivity and limited bedrock incision? Did you find that groundwater-surface water connectivity through discrete fractures was highly variable in space and time, and depended on fracture size or aperture, river stage, and the distribution of hydraulic head within the flow system?

Response: We agree with the reviewer. We will provide a clearer set of conclusions based on the questions posed in a revised introduction. A similar comment was raised by Reviewer 1.

There is a huge amount of data contained in this manuscript, however in some cases data was left unused in interpretations and discussion. For example, precipitation & snowfall, daily river stage, fracture content as a function of depth, atmospheric temperature, etc. Why include these data if they are not utilized? In the end, if the hypothesis was to test "will ERT work for this" I think that was not clearly answered, and furthermore, given the high dependence on temperature, it may be that that answer is "no."

Response: These data provide critical context for the reader; they also support elements of the conceptual model (e.g., fracture networks in the rock). The hydrology data summarized in Figure 4 provides critical context for the geophysical observations and discussion of seasonal trends. Our discussion and interpretation of seasonal transients in the geophysical measurements implicitly utilizes the hydrological information. Without these data the reader would not be able to assess the severity of the seasonal temperatures fluctuations and river stage (presented here as flow), precipitation and snowfall on the geophysical response, or appreciate the frequency of our measurements within spectrum of field conditions. These data serve as a critical point of reference or comparison for other investigations in rivers. Further, the exclusion of these data would not improve the manuscript nor would it reduce the number of figures, thus, we support the inclusion of these data.

Nevertheless, the specific contributions of this study to the conceptual model will be more clearly stated. One of those contributions is the strong influence of temperature, which is viewed as an indicator of the vertical extent of surface water influence (direct or indirect) within the bedrock riverbed.

Specific Comments:

Line 55: Perhaps add a comment on what Fan et al., 2007 found here?

Response: Clarifications will be implemented.

Line 63: There is a lot going on in the figure and it is weakly linked to the text. Are you testing these concepts?

Response: The conceptual model is meant to represent the system in its entirety. We agree that the figure was not as strongly linked to the text as it should have been. We believe this can be addressed. It is important to present the conceptual model and elements that, in theory, could be explored or tested with the approach used in this study. There are, however, limitations to our study. The figure serves as motivation or starting point to our conceptual understanding of groundwater-surface water interaction or hyporheic zones in fractured rock. It is a simplified view of a fractured rock system, showing advection and diffusion processes for a gaining and losing stream; these are well-known concepts in sedimentary fractured rock. We do believe that the text can be improved to better link our study to the elements of the conceptual model.

Line 92: The Singha paper has 2014 printed on top of it, but I'm not sure which date is correct.

Response: It appears to be published on-line in 2014. But the paper wasn't fully published until 2015.

Line 175 – 180: Was the formation of basal ice actually observed at the site or only inferred?

Response: Basal ice was visually observed in the field as indicated in the manuscript. The ice was no longer visible once the river froze over.

Line 213 – 216: I am unfamiliar with this method of sampling temperature while the sensor is in motion. Certainly the sensor itself, however small it might be, has some thermal mass that requires time to equilibrate to the surrounding water temperature. Even though the sensor is capable of measuring at 0.5 s rate, that does not mean that the measured data are reflecting changes in the formation at that rate. A reliable reference should be included here to justify the method, and a controlled validation test and sensor calibration under laboratory conditions should be conducted to quantify sensor response.

Response: This is very interesting question and one that we had not considered until now. Based on the information we received from the RBR*solo*[TM] manufacturer (RBR Limited, Ottawa, Canada), these sensors will resolve 63% of a full-scale temperature change in 1 s, 95% in 1.5 s and 99% in 2 s. This particular sensor has a maximum resolution of 0.00005°C (full scale). However, in this study we only report temperature changes to the 0.01°C. Given our reported rate of decent ~0.8 cm/s combined with our temperature resolution, a conservative vertical "averaging" estimate might be ~1.5 cm based on the full resolution. Therefore, sensor response time in this case appears to be very small and negligible on the data sets presented. Unfortunately, we do not have the laboratory equipment to make any further comments on the performance of this sensor deployed in this way.

We can include some of this information in a revised manuscript.

Line 219: What is the value of measuring snowfall accumulation if snow density is not also reported with a conversion to SWE?

Response: Snow accumulation is reported as SWE. Clarification of SWE will be provided in the figure caption.

Line 221/Figure 4: What scale are the red dot "Resistivity Samples" on? They appear to be only temporal and unitless, however they seem to track the river stage which is confusing. Is 'snowfall' in this figure converted to SWE? If not, please do and clarify the label.

Response: The resistive samples "red dots" are plotted to show their temporal position and sampling interval during the seasonal hydrological conditions of the river. They track the river stage (y-axis) because these measurements were collected in the river and the stage is explicitly used in the models. These red dots effectively identify the stage conditions sampled in this study.

Line 228: "effective sensing depth" does this mean measurements are reflective of the 6m depth zone, or the entire aggregated zone 0 to 6m depth?

Response: The instruments sampling depth (volume) is defined by the impulse response function (McNeil, 1980). Here, the sampling depth is stated as 6 m which is general rule-of-thumb to the depth of investigation for the instrument in this orientation. More descriptive phrasing can be added to the text.

Line 237: "BLANKED by bedrock rubble" I am not familiar with this usage of "blanked" in this context. Suggest rewording for clarity.

Response: The word was "blanketed" but we can change this to "covered" to avoid confusion.

Line 239 – 244: Does the electrode construction method have any particular importance to this study? This sounds like very typical ERT cable construction, albeit by the end-user rather then a professional fabricator. Probably could be deleted.

Response: In theory the construction of our cables is similar to commercial systems, but because it isn't a commercially available cable we felt it best include the details of its construction. There are design elements that could have an impact on the results (e.g., electrode construction and length) that a reader might want to know. Our inclusion of the design of our cable is consistent with the approach of other workers (e.g., Van Dam et al. 2014).

Line 261: How was the measurement time determined? It is known that diurnal fluctuations in stream water temperature may be of magnitude in excess of 10C, similar to your annual range of groundwater temperatures. Also you acknowledge the affect of temperature on the ERT readings; how does the timing of the measurements affect the data due to daily fluctuations?

Response: This is a great question. Although we discuss the limitations of this study (data aliasing) and potential sources of thermal influence on the geophysical measurements, we do not neglect temperature fluctuations. These are considered to be an important component of the observed geophysical dynamics. Our sampling frequency considers longer-period (seasonal) variations rather than diurnal variations. Given our coarser sampling interval we cannot comment on the impacts of shorter-period temperature fluctuations (diurnal) on the groundwater resistivity. We discuss the potential impact of sunlight (heating) of the riverbed and its spatial/temporal variability on the geophysical signatures in the last paragraph of Section 5.1.

Further clarification on this subject can be incorporated as well as the inclusion of the references provided by the reviewer below.

Constantz, Jim, Carole L. Thomas, and Gary Zellweger. "Influence of diurnal variations in stream temperature on streamflow loss and groundwater recharge." Water resources research 30.12 (1994): 3253-3264.

Constantz, Jim. "Interaction between stream temperature, streamflow, and groundwater exchanges in alpine streams." Water resources research 34.7 (1998): 1609-1615.

Line 262: "manually filtered" What criteria was used for manually filtering? Why was this approach used rather than the common quantitative method of envelope filtering based on an error model?

Response: In our case, only obvious data outliers were removed (e.g., failed measurement based on a non-zero standard deviation); we intentionally did not apply any pre-inversion data smoothing or averaging in an attempt to preserve the data trends and maintain data-input consistency. However, data smoothing was direct applied in the inverse routine as described in the text. The approach used in study will depend on the site conditions and desired outcome of the experiment. In our case, we were concerned with preserving the signal of the natural system (governed by multiple factors) rather than enhancing a particular element or physical processes in the model. Additional details of the inversion setup will be added to the manuscript.

Slater, Lee, et al. "Cross-hole electrical imaging of a controlled saline tracer injection." Journal of applied geophysics 44.2 (2000): 85-102.

Line 264: "moderate to high damping" – Does this mean different damping factors were used on each dataset? What is the numerical value of damping used and how is this value incorporated into your inversion scheme?

Response: The initial damping factor was moderate but allowed to vary in the inverse routine; the same starting parameters were used for each model run. However, models were allowed to optimize the dampening factor depending on the model convergence. Therefore, the dampening factor likely increased with noisier datasets. Additional details including specific parameters used in the routine can be added to the text.

Line 267: What parameters, how were they optimized, and were identical settings used for all datasets?

Response: Additional details and clarification will be incorporated into the text.

Line 268 – 269: Certainly achieving the lowest possible RMS is not the optimal approach to achieve the most "believable" geophysical result. How does the RMS relate to observed measurement noise/errors? At what point is the inversion fitting noise?

Response: We agreed. That being said, minimizing RMS error (minimization between measured and modeled data) is a reasonable approach to achieving the most representative model. Of course, the model is only as good as the measurements inputted into the model. One of the challenges in this study was finding a reasonable solution to the data collected in the winter (frozen) months. These periods were accompanied by erroneous data points (largely due to the high contact resistances of frozen ground and ice) higher overall noise, and thus, model convergence and stability was at times challenging. The issue could only be circumvented by imposing a maximum limit on modelled resistivity. This meant that the

model RMS errors would be considerably higher (note the data presented in Figure 10). Therefore, minimizing the RMS errors was not the only criteria used to achieve the most realistic model.

Line 280: Is there a reference for this Resistivity Index? What is the justification for manipulating the data in this way?

Response: The resistivity index (a broadly and routinely used normalization technique) was used here so that we could compare the transience observed between the pool and riffle sections, which had very different mean resistivities. The index simply allowed us to normalize the data sets to the mean value, permitting easier comparison of the timing and magnitude of transients at each site. Our approach is defined in Equation 3. We can also provide a better justification for its use in the methods section.

Line 293 – 294: Where is the data demonstrating upward head shown?

Response: The data was not included because it was not used in the study other than for the purpose of establishing the direction of potential groundwater flow. Since the data did not reveal any major changes in the gradient we decided not to include the data, and instead, simply state (in words) what the data showed. The hydraulic head data could be added to Figure 7.

Line 307/Figure 6b: It would be helpful to grade the colors of the lines linearly to more easily show the temperature trend. Even better would be to present these data as a matrix/grid where time is on the x-axis, depth is on the y-axis, and color represents temperature.

Response: This is a good recommendation and one that we will consider in the future. However, we think the current presentation is also reasonable and maybe more "understandable" at a quick glance. The purpose of the figure was to show the temperature swings (dynamics) with respect to depth, and the extent of the heterothermic zone, and illustrate the temporal variability overserved during the winter months. We think the current figure layout achieves these objectives.

Line 308: "correspond to areas" I cannot tell from the figure how the fracture patterns correspond to the temperature results. Perhaps some annotation, or another approach to presenting these data would help.

Response: We agree that some annotation to the Figure 6a should be added to better illustrate the position of fracture zones.

Line 321 – 331: [Figure 8] This seems more like a discussion point rather than a result.

Response: We agree. This sentence would be better placed in the discussion or conclusion section.

Line 356 – 358: "greater number of measurements. . ." why would the number of removed data cause higher RMS? Presumably if the data were removed, they would no longer be included in the RMS calculation.

Response: The missing qualifier in our text is that while obvious outliers were removed, the overall dataset remained nosier compared to unfrozen periods. Further explanation is provided above (see comments for Line 268 – 269), and necessary revisions to the text can be made.

Line 374: What are the observed thicknesses of basal ice and floating ice?

Response: Unfortunately we were not able to measure the thickness of basal ice. It was slimily visually noted in the field.

Line 415-416: "groundwater discharge in this section" I don't follow the logic why the relationship between substrate resistivity and "surface water response" indicates magnitude of discharge that could be interpreted in this way. Also, correlation is not demonstrated or quantified.

Response: Here, we are simply saying the transience observed in surface water (days 13, 26, 31; Figure 7) are not readily apparent in the riverbed, and thus, surface water may not be interacting with groundwater during these periods. This suggests that groundwater discharge may be a more dominant/overriding process. We can modify the text to improve the clarity of this statement.

In this case, "correlation" is not accompanied by a statistical qualifier; therefore, it should be read as qualitative description of a relationship between to things.

Line 432: "strong upward hydraulic gradients" please indicate where this is demonstrated by data.

Response: There are two options: we can include these data in Figure 7 or simply state the gradient range in the text. Given the limited use of the data we will include the calculated gradient range in the main body.

Line 436: "likely dominated the bulk electrical response" Why 'likely'? Based on the evidence shown, temp is clearly dominating the ERT signal.

Response: Agreed. This can be changed.

Line 445: Where is ground frost or riverbed ice formation measured data shown?

Response: Neither ground frost or river bed ice were explicitly measured in this study. Ground frost was interpreted based on the resistivity data (Figure 11e), while riverbed ice (basal ice) was observed in the field as described in the text.

Line 459 – 473: As previously stated, data showing the frost and ice should be shown.

Response: Please see response above.

Line 474: I'm not sure what evidence directly supports this statement. The provided sensitivity analysis appears to only vary the river water electrical properties; this doesn't seem to directly simulate the presence of ice as claimed in this statement.

Response: The reviewer comments are understandable. We did not mean to suggest the "sensitivity analysis" was done to evaluate the influence of river ice. We can address this issue with some reorganisation of the text.

Line 479 – 480: Quantify this? Why would inputting a one-half of true river water resistivity lead to "substantial overestimates of river resistivity" – wouldn't the river water resistivity be fixed so that the output = the input?

Response: We were referring to the implications of fixing the model with an inaccurate water resistivity. The point of the statement is to highlight the importance of accurately representing the geometry and resistivity of the surface water in the inverse model; accurate surface water information was inputted into our models with the possible exception of frozen periods where river ice could have altered the geometry of the surface water layer.

Line 486: What about a synthetic model example?

Response: At this point we will limit our results and discussion to the field measurements. A synthetic study would be very informative and could be considered in future work.

Line 510: How does geoelectrical transience translate into hydrological processes?

Response: Changes in electrical resistivity of an area over time indicate variations in the electrical properties of the pore water (the only dynamic component of the system). Changes can occur as a result of temperature, specific conductance, or saturation. Surface water and groundwater typically exhibit distinct electrical properties (Figure 8). Whether these properties can be exploited in bedrock environment using surface geophysical methods has not previously been explored. Our study aims to assess the utility of surface geophysics, while also examining the utility of temperature and EC fluctuations to infer hydrological processes (e.g., thermal conduction, groundwater-surface water exchange, and fracture connectivity) in a bedrock river of varying morphologic conditions.

Line 511 – 452: The conclusions section contains substantial summary and could be reworked for improved focus.

Response: We agree with this assessment. We will rework the conclusions to better highlight the specific contributions of this study with respect to the method and application in bedrock river environments, as well as elements of the conceptual model described in the introduction.

Figure 5: The purpose of this figure is unclear and I suggest that it could be deleted. The A/B/C/D locations are already indicated on Figure 12; the river stage information is presented on Figure 4; the location of the model block midpoints does not appear to be substantially important to the manuscript.

Response: We agree that Figure 5 is not necessary.

Figure 9: Perhaps showing only the difference between these two maps would make interpretation easier? If not only difference, then perhaps just adding a third difference panel. Also, isocontour labels are too small to read.

Response: This figure can be modified to also show the change or difference between data sets. The labels on the contours can be increased in size.

Figure 10: What is the model error relative to the measurement errors? What is the purpose of showing these vast bulk averages when that eliminates any of the valuable spatial information yielded by using tomographic methods? Figure 12 seems to be much more useful than this.

Response: Quantification of measurement error is not straightforward. A precise measurement is not the same thing as an accurate measurement. It is generally easy to obtain a precise measurement by stacking. Imprecise measurements are negated as outliers. However, the accuracy of the measurement has more to do with site conditions (e.g., heterogeneities, anisotropy). Our use of a Wenner array (selected for its higher signal quality) limited our ability to assess the accuracy of the measurements. Other arrays, like the dipole-dipole, permit the collection of reciprocal data, which can be used to quantify measurement error.

The primary purpose of this figure was to the present the time series in its entirety for each location, including the min/max/median value, model RMS error, data removed and relative position of the selected resistivity snapshot shown in Figure 11. The figure also summarizes the field conditions for each

measurement (e.g., unfrozen, partially frozen, frozen) which is important for the interpretation of data in Figure 11 and Figure 12. Figure 12 does not provide this information.

Figure 11: Very nice layout and presentation of this figure, however certainly this needs to be replotted to show the difference between (b) through (h) relative to (a) in both columns

Response: This comment is understandable; however; we did not apply a time-lapse inversion due to the large temporal sampling interval and varying data quality (removed data points) over the course of the study, both of which degrade value of time-lapse inversions. In our opinion, presenting the data as absolute resistivity f (rather than relative % change in resistivity) is more representative of the site conditions. At this point we would prefer to maintain the current layout. Figure 12 provides an indication of the relative change in resistivity for specific region in the model at each site.

*References:*

*McNeill, J. D.: Electromagnetic terrain conductivity measurement at low induction numbers, Geonics Ltd. Technical Note, TN-6, 1980.*

*Van Dam, R.L., B.P. Eustice, D.W. Hyndman, W.W. Wood and C.T. Simmons: Electrical imaging and fluid modeling of convective fingering in a shallow water-table aquifer, Water Resources Research, 50, doi: 10.1002/2013WR013673.*

---

## Author Response (AR1)

**Author Response to Reviewer Comments**
**Hydrol. Earth Syst. Sci. Discuss., doi:10.5194/hess-2016-559, 2016.**

First and foremost, we would like to thank the reviewers for donating their time to review our manuscript and for providing a fair and constructive review of our work. Our response to general and specific comments is documented below, and reflected in a tracked-version of the revised manuscript.

Based on the general and specific comments below, we focused on improving the introduction, discussion and conclusions. We have revised our problem statement and provided a clearer definition of the scientific objectives as they pertain to the general conceptual model and the style of our geophysical measurements. Our discussion of the geophysical results and its significance to the hydrologic community has been improved by better integrating our interpretation of the various data sets, thereby strengthening our general scientific contributions in the conclusion. Individual reviewer comments have been addressed in the manuscript where necessary. Although some of the technical issues associated with the source of measurement errors in our resistivity method remain an inherent limitation in our survey design, we believe that our approach was reasonable given the site conditions and research objectives; we have provided a thorough explanation of all comments below.

All author responses refer to the revised tracked version of the manuscript, while reviewer comments (i.e., line numbers) refer to the original submission.
* * *
**Reviewer 1: General Comments**

The study deals with fractured sedimentary bedrock riverbeds and its spatio-temporal groundwater surface water exchange of the Eramosa River within the Grand River Watershed, Ontario, Canada. Surface electrical methods (ERT and EMI) were used for a quasi non-invasive assessment of the scale and temporal variability of riverbed temperature and groundwater surface water exchange beneath its sedimentary bedrock riverbed. Underpinned were the solid geophysical data sets by a network of boreholes and streambed piezometers, installed across the site. The study further contained highly relevant material for HESS, is well written, structured and citied and attested the authors' fundamental background and experiences with the selected topic. Fractured sedimentary bedrock systems and its interaction processes are very complex and difficult to describe. Hence the presented study provided a useful approach for cost-efficient investigation into the river flow regime. Still there are a few issues that needs to be addressed until the manuscript is ready for publication. Although the study impressed be the amount of high quality data and different applied methods, it lacks a little of a clear and comprehensive purpose.

Response: We have revised the introduction to highlight the state of the science, identify the specific research needs, and outline the specific contributions of our study adds to the conceptual model. We also incorporated more specific details regarding the regional groundwater flow system.

Studies of interaction flow processes at rivers are relatively common since about one decade; however, the presented geophysical investigation of bedrock rivers combined with ground truths core data is unique. Moreover, the economic and ecological potential of global bedrock rivers are remarkable, hence the study can be considered as a pioneer study with portable conclusions. This seems the strengths of this study, however it needs to be emphasized.

Response: We have provided a more detailed summary of the unique contributions of this study in the introduction (Lines 183-196) and its relevance to a broader environmental audience. A similar emphasis was implemented in the final paragraph of the conclusion section.

It would be helpful if the authors establish somehow a relation between their investigations and findings, and a direct affected related system, such as a water supply or ecosystem services. If not available for the selected test site then relevant literature can be used. In this respect the manuscript might be slightly rearranged. Introduction should contain a clear state of the art containing the situation, the problem, the challenge and the provided response (solution).

Response: The basic conceptual model of a groundwater-surface water mixing zone in a fractured bedrock river is now presented in the introduction. The mechanisms controlling this exchange are explained using the conceptual model (Fig. 1) in light of related literature. This is followed by an explanation of our study approach and motivation, along with the specific challenges and the anticipated contributions of this work. Refer to Lines 142-196.

Although I like the extra paragraph 2 'background', it's quite uncommon and could be incorporated into the introduction and the material and method part, respectably. It is always helpful to read what other studies have archived at similar test sites and / or with similar approaches and where they were limited. This should be highlighted in the introduction and in the conclusion (and in the abstract too).

Response: We have re-arranged the background section. The literature review is now presented in the introduction (Lines 82-141).

The conclusion should hence contain the extracted information given by the findings as kind of a 'take home message' for the scientific community rather than a second abstract. I recommend acceptance with minor revision.

Response: We have revised the conclusion section to better highlight the specific conclusions of this study. The first paragraph provides an overview of the work. The second and third paragraphs provide the specific contributions. The fourth and final paragraph explains the impact and relevance of this study to a broader scientific community.

**Specific Comments:**

Line 94 – 96: amount of references can be decreased by a related review

Response: We have maintained the original citation list.

Line 126: please mention that the presented Archie's law is simplified are provide the whole equation

Response: We have added "a fluid saturated medium" to identify the simplified form of this equation (Lines 498). We did provide additional text regarding the underlying assumptions of this equation (Lines 503-506).

Line 144: just mentioned the temperature correction was done by Arps (1953) might be sufficient

Response: Since this equation was used to assess the influence of temperature on the bulk formation resistivity we think it would be best to provide the equation. This equation and associated discussion is now provided in the Methods section (Line 513-522).

Line 224: indicate the approx. location of the EMI lines in Figure 2

Response: Figure 2 has been updated to show the extent of the EMI survey area.

Line 270: use 'Fig' or Figure either but consisted

Response: The full word should be used at the beginning of a sentence but abbreviated in-line.

Line 291: matrix porosities from the corehole relatively low in respect to? A short reference value for the same rock material from the literature is useful or do you mean in comparison to the weathered or broken rubble zone? If so, please mention it

Response: For simplicity we have removed the phrase "…were relatively low" as we do not compare these values to other samples.

Line 321: see above

Response: No further response.

Line 328: it is hard to follow here or maybe I missed the point how you ended up with the 46%, according to Archie in Eq.1? How do you get the Sigma w values or am I totally off? If you re-arrange the equation it needs to be mentioned if not showed

Response: We have modified the wording in the body (Lines 737-769) and the figure caption (Lines 1576-1583). Figure 7 shows the range in formation resistivity based on measured pore water EC (the plot shows effects specific conductance variations and the calculated influence of temperature). Given that water temperature is dynamic and will influence the formation resistivity, we have also provided the potential effect of temperature fluctuations on the formation resistivity. This effect is shown by the bounding isotherms for both groundwater and surface water. The % change in formation resistivity represents the maximum change based on the potential temperature range (e.g., 4-14°C or 0-18°C).

Line 332 – 349: how could you be sure that the EMI data were not affected by outside conditions, do you temperature corrected the ECa as well?

Response: Assuming the reviewer is referring to atmospheric temperature effects on the instruments performance, we do not believe, nor does the manufacturer report that ECa measurement accuracy of the EM-31 (+/- 5% at 20 mS/m) would have any meaningful effect on the reading. The ECa measurements we report in Figure 8 are, of course, sensitive to the temperature of the media as indicated by Eqn 2 and illustrated in Figure 7. ECa variations in the rock are expected to be associated with either temperature fluctuation or specific conductance. Temperature fluctuations within the rock have be used as a tracer to map a zone of influence or transience below the riverbed by other workers. The differences identified in this study could be associated with changes in seasonal groundwater-surface temperature, mixing or changes in groundwater chemistry. The data we present identifies measurable changes in the geoelectrical conditions beneath the riverbed, whether it's due to temperature, EC or some combination of the two; regardless, the spatial extent of these dynamics could support the existence of a mixing zone or simply a zone of thermal variability in the rock unique to this type of river.

Line 420: I prefer including of the discussion together with the presentation of the results. This helps to shorten the manuscript which is almost every time helpful

Response: We appreciate the reviewer's suggestion. Considering the nature of the discussion in this paper we felt it best to separate it from the results to avoid potential confusion (i.e., geophysical results vs. hydrogeophysical interpretation). At this point we have maintained a separate results and discussion section.

Line 857: since you presented the ERT results in Figure 11 by common scale, Figure 10 is kind redundant, mention in the text that the ERT data quality (RMSE, removed data points) were higher under frozen, partly frozen conditions

Response: While we understand the reviewer's comment, we believe the information presented in Fig. 10 is highly informative and critical to the interpretation of the selected data sets in the subsequent figures. The study does contain an immense amount of information, which cannot be presented or summarized in its entirety, but the seasonal trends need to be captured so that reader can fully appreciate the highlighted information. The unique field conditions resulted in many challenges in data processing and interpretation that are best emphasized in Figure 10 rather than hidden or ignored entirely. To our knowledge, there has been no similar study performed in a river in this type of environment, let alone a bedrock rock river, so we think it is critical to provide as much of the data, as possible, particularly if they provide insights into the modelling results.
* * *
**Reviewer 2: General Comments**

This manuscript is on the topic of geophysical and traditional measurements of a reach of river to investigate the suitability for time lapse ERT to study river bottom processes. The writing is in clear, good English, and the figures are mostly very readable and nicely drafted. The topic – either from the Hydrology or Geophysics perspective – certainly has the potential to be of interest to HESS readership. I believe the topic of this work fits into the scope of this journal. The most significant limitation I see to this work is related to the experimental design, which is largely absent from the writing. In short, it is difficult to tell what was being tested about the hydrology, and why measurements were implemented to carry out that test. The stated hypothesis is apparently related to "will the geophysics work," while the theme of riverbed processes appears and disappears throughout the manuscript. In the end, I remained confused about exactly what the reader was meant to take away from this given the setup of the writing and the design of the study. There is certainly lots of good data here and on some level this has the potential to be of high interest to the hydrology community, but there is a need for substantial revision for focus.

Response: We very much appreciated the reviewers' assessment here. We have made substantial revisions to the introduction, specifically relating to the basic conceptual model (motivation) and the examination of mechanisms controlling groundwater-surface water exchange (hypotheses). The model is now presented in light of existing literature to show the value of our study and provide a clearer description of our study design and elements of the conceptual model we aim to address. The objectives of this study have been re-emphasized and are now more specific to the experimental design/style of measurements collected (Refer to Lines 142-182). Similar changes to the text were implemented in the Conclusion section.

There were several other notable issues/limitations related to measurement methods, data processing, and absence of some measurements that are detailed in my General and Line-by-line comments below. At this time I am recommending this manuscript be returned for major revision, however if the experimental design is not substantially clarified and the focus reworked to highlight hydrological interpretations, a second review would likely not result in a favorable recommendation.

There is a substantial disconnect between the topic of the science question and the posed hypotheses. Although the science question is not explicitly stated, it is my best interpretation that the following reflects the intent of inquiry: ". . . there remain gaps in our conceptual understanding of groundwater-surface water interaction and exchange mechanisms in bedrock rivers where discrete fracture networks will dominate groundwater-surface water flux with secondary interactions supported by the porous rock matrix." On the other hand, the hypothesis is explicitly stated, although it appears to be limited to a yes-or-no "will it work" type of speculation: "we hypothesize that a groundwater-surface water mixing zone – encompassing fracture and matrix flow and diffusion – may be identified within a fractured bedrock riverbed by monitoring spatiotemporal changes groundwater temperature and porewater electrical conductivity using minimally invasive electrical resistivity methods." Further complicating matters is the text between Line 69 and 76 that highlight the hydrological outcomes while disregarding the stated hypothesis.

Response: We acknowledge that there are inherent limitations in our study some of which could have been addressed given the benefit of hindsight. We also understand that this study represents a step toward the advancement of the conceptual model. We describe the physical processes with a simple conceptual model based on existing literature from alluvial river studies and fractured sedimentary rock. Our hypothesis was that surface geophysical measurements, specifically electrical methods, may be used to detect changes in geoelectrical properties beneath a bedrock riverbed; this hypothesis was derived from the success of similar studies in alluvial rivers, yet it has never been tested. Therefore, we designed a long-term seasonal resistivity monitoring program, complemented by continuous measurements of groundwater-surface water temperature and EC, to test this hypothesis. In our case, we were able to explore a range of seasonal conditions (including freeze-thaw) and their impact on the bulk electrical properties of the rock. Given that no previous geophysical (or hydrological) studies had been conducted in a bedrock river we did not know *a priori* the effectiveness of our geophysical approach particularly in a hydrological context. In the end, the experiment was successful from the perspective of appreciating the capabilities and limitations of these geophysical methods. Given the benefit of hindsight, we acknowledge that our experiment could have been designed differently or in a way that would have allowed us to examine a different set of questions, such as source of measurement errors and their impacts on the inverse models. That being said, we believe our approach was reasonable given our primary desire to achieve the highest signal-to-noise ratio. Further, comments on this topic are provided below.

Throughout the manuscript, speculative statements about river ice, river-bottom ice and frost are made, though they do not appear to be supported by any direct measurements or observations. Estimates of loss along reaches based on calculations of discharge using rating curves in conjunction with stage height monitoring appear to be absent. This line of evidence would substantially help to support geophysical observations.

Response: We have improved the clarity of our statements regarding river and basal ice. While these features were not directly measured they were observed and did have an impact on our results and interpretation. We also acknowledge the value of monitoring river loss-gain along the reach as this would been helpful in understanding local areas of discharge or recharge in relation to the geophysical measurements; unfortunately this information was not collected in a way (i.e., period and sampling frequency) that would be useful in a direct way to interpret the geophysical dynamics. However, we would like to point out that this data was collected and will be examined in a future paper, at which point, it can be compared to the results of this study.

I felt that the following questions posed early on in the manuscript were not clearly answered: Do you find that groundwater-surface water interaction was restricted by poor vertical connectivity and limited bedrock incision? Did you find that groundwater-surface water connectivity through discrete fractures was highly variable in space and time, and depended on fracture size or aperture, river stage, and the distribution of hydraulic head within the flow system?

Response: We have revised the questions posed in the introduction to be more representative and consistent with the measurements and observations in this experiment. While these fundamental questions regarding the nature of groundwater-surface water remain, our conclusions are now more reserved given the inherent limitation of geophysical measurements, and absence of direct hydrological information such as groundwater discharge and recharge along the river. The nature of groundwater temperature fluctuations beneath the riverbed remains a central theme of this work.

There is a huge amount of data contained in this manuscript, however in some cases data was left unused in interpretations and discussion. For example, precipitation & snowfall, daily river stage, fracture content as a function of depth, atmospheric temperature, etc. Why include these data if they are not utilized? In the end, if the hypothesis was to test "will ERT work for this" I think that was not clearly answered, and furthermore, given the high dependence on temperature, it may be that that answer is "no."

Response: Precipitation and weather data provide critical context for the reader and the geophysical measurements we discuss; they also support elements of the conceptual model including thermal zones of influence and the fact that groundwater and surface water can reside in different temperature regimes. Our discussion and interpretation of seasonal transients in the geophysical measurements implicitly utilizes the hydrological information from the perspective of seasonal temperature, precipitation events and river stage fluctuations. We have implemented a number of revisions throughout the manuscript to strengthen the link between atmospheric information and the geophysical measurements.

It should be noted that temperature was viewed as a possible tracer in the delineation of a groundwater-surface water mixing zone; this was motivated from an extensive body of literature utilizing temperature fluctuations to estimate groundwater discharge in rivers (e.g., Lines 121-127). One of our primary goals was to evaluate the relative magnitude of temperature effects on formation resistivity relative to other parameters such as specific conductance and phase transformation. In this case, our study was very successful in identifying the dominance and spatial extent of seasonal temperature transience beneath the riverbed. The mechanisms governing those transience (e.g., fracture aperture, connectivity etc.) will be examined in future work, and will ideally capitalize on the results of this study.

Specific Comments:

Line 55: Perhaps add a comment on what Fan et al., 2007 found here?

Response: Additional details have been added on Lines 76-79.

Line 63: There is a lot going on in the figure and it is weakly linked to the text. Are you testing these concepts?

Response: The conceptual Figure 1 serves as a point of reference and source of motivation for this study. It was not our intent to explicitly test and confirm the various elements described in the model, but rather to illustrate their existence and likely contribution to geoelectrical transience beneath the riverbed. We have improved the placement of the conceptual model in the instruction along with pertinent literature and discussion; we also provide a better description of its relevance to this study. The basic concept we wanted to test was whether or not there was an exploitable geophysical signature, and explore the factors or mechanisms controlling these responses. This would provide at first-order understanding of the geoelectrical response and seasonal transience characteristic of fractured bedrock rivers.

Line 92: The Singha paper has 2014 printed on top of it, but I'm not sure which date is correct.

Response: Based on our records the paper was published on-line in 2014 but wasn't fully published until 2015.

Line 175 – 180: Was the formation of basal ice actually observed at the site or only inferred?

Response: Basal ice was visually observed in the field as indicated in the manuscript. The ice was no longer visible once the river froze over. We have modified the text throughout to clarify this point.

Line 213 – 216: I am unfamiliar with this method of sampling temperature while the sensor is in motion. Certainly the sensor itself, however small it might be, has some thermal mass that requires time to equilibrate to the surrounding water temperature. Even though the sensor is capable of measuring at 0.5 s rate, that does not mean that the measured data are reflecting changes in the formation at that rate. A reliable reference should be included here to justify the method, and a controlled validation test and sensor calibration under laboratory conditions should be conducted to quantify sensor response.

Response: This temperature trolling method has been published by other workers (e.g., Pehme et al. 2010; Pehme et al. 2014, Lines 717-721). However, in this study we utilize a different sensor deployed in a slightly different fashion. The basic approach and resulting data sets are the same.

With respect to the sensor's sampling frequency we can say that these RBR*solo*$^{TM}$ sensors (RBR Limited, Ottawa, Canada) resolve 63% of a "full-scale" temperature change in 1 s, 95% in 1.5 s and 99% in 2 s. This particular sensor has a maximum resolution of 0.00005°C (full-scale) based on our communications with the manufacture. In this study, we report temperature changes to the nearest 0.01°C. Given our reported rate of decent ~0.8 cm/s combined with our temperature resolution, a conservative vertical "averaging" estimate might be ~1.5 cm based on the full resolution. Therefore, sensor response time in this case appears to be very small and negligible on the data sets presented. Unfortunately, we do not have the laboratory equipment to make any further comments on the performance of this sensor deployed in this way.

We have incorporated additional details on Lines 569-570 to explain the sensors sensitivity and response time.

Line 219: What is the value of measuring snowfall accumulation if snow density is not also reported with a conversion to SWE?

Response: Snow accumulation is reported as SWE. Clarification of SWE has been incorporated into the figure caption on Line 1548-1549.

Line 221/Figure 4: What scale are the red dot "Resistivity Samples" on? They appear to be only temporal and unitless, however they seem to track the river stage which is confusing. Is 'snowfall" in this figure converted to SWE? If not, please do and clarify the label.

Response: The resistive samples "red dots" are plotted in Figure 4 to show their temporal position and sampling interval during the seasonal hydrological conditions. In this case, they track the river stage (y-axis) because these measurements were collected in the river and the stage is explicitly used in the models. Although this could be viewed as unconventional we think this is easiest way to present the data. We have revised the caption of Figure 4 to resolve any potential confusion on the meaning of the red dots.

Line 228: "effective sensing depth" does this mean measurements are reflective of the 6m depth zone, or the entire aggregated zone 0 to 6m depth?

Response: The instruments sampling depth (volume) is defined by a non-linear impulse response function; the sampling depth or interval is dependent on the coil separation and orientation of the induced field. This information can be found in McNeil (1980). Here, the sampling depth is stated as 6 m which is a general rule-of-thumb to the depth of investigation for the instrument in this orientation. We have added "cumulative depth" on Line 588.

Line 237: "BLANKED by bedrock rubble" I am not familiar with this usage of "blanked" in this context. Suggest rewording for clarity.

Response: The word was "blanketed" but we have changed the word to "covered" to avoid confusion.

Line 239 – 244: Does the electrode construction method have any particular importance to this study? This sounds like very typical ERT cable construction, albeit by the end-user rather then a professional fabricator. Probably could be deleted.

Response: In theory the construction of our cables is similar to commercial systems, but because it isn't a commercially available cable we felt it was best to include the details of its construction; some of the design element are unique, which some readers may want to know. Our inclusion of the design of our cable is consistent with the approach of other workers (e.g., Van Dam et al. 2014, Water Resources Research).

Line 261: How was the measurement time determined? It is known that diurnal fluctuations in stream water temperature may be of magnitude in excess of 10C, similar to your annual range of groundwater temperatures. Also you acknowledge the affect of temperature on the ERT readings; how does the timing of the measurements affect the data due to daily fluctuations?

Response: In this case, measurements were collected as frequently as possible; however the main goal was to track larger period fluctuations or seasonal changes in the environment. Given the larger spatial scale of the resistivity measurement and interpretation of early results we knew that measurable contrasts were most likely to occur through seasonal transitions. We did attempt to capture short-period transience, particularly during the second half of the study, which can be seen in the denser sampling interval in Figure 4.

Although we discuss the limitations of this study (data aliasing) and potential sources of thermal influence on the geophysical measurements, we do not neglect temperature fluctuations. These are considered to be an important component of the observed geophysical dynamics. Our sampling frequency considers longer-period (seasonal) variations rather than diurnal variations. Given this coarser sampling interval we cannot comment on the impacts of shorter-period diurnal temperature fluctuations. That being said, we did acknowledge the potential impact of sunlight (heating) of the riverbed and its spatial/temporal variability on the geophysical signatures, and thus, the timing of the measurement. We have enhanced this discussion with additional text and references (i.e., Constantz et al. 1994; Constantz 1998) on Lines 986-1012.

Line 262: "manually filtered" What criteria was used for manually filtering? Why was this approach used rather than the common quantitative method of envelope filtering based on an error model?

Response: In our case, only obvious data outliers were removed (e.g., failed measurements based on a non-zero standard deviation); we intentionally did not apply any pre-inversion data smoothing or averaging in an attempt to preserve the data trends and maintain data-input consistency. However, data smoothing was directly applied in the inverse routine as described in the text. The approach used in any study will depend on the site conditions and desired outcome of the experiment. In our case, we were concerned with preserving the signal of the natural system (governed by multiple factors) rather than enhancing a particular element or physical processes in the model.

Measurement error is indeed a very important factor in the interpretation of resistivity data. Noise can arise from low signal (S/N ratio) such as electrode contact or instrumentation capabilities. These sources of noise are dynamic and minimized through data stacking in the field. Alternative sources of noise can arise from localized heterogeneities or non-uniform ground coupling, effectively resulting in the creation of artifacts in the data (2D solution to a 3D problem). Our filtering approach removed stacked measurements with a standard deviation exceeding a couple %; in most cases these data points were easily identified based a sudden/localized fluctuation in resistivity. However, the other and likely more significant source of error, arising from spatial heterogeneity, was not easily quantified due to our intentional use of a Wenner array. Although this array yields a higher S/N ratio it does not permit the collection of reciprocal data. Slater et al. 2000, Journal of Applied Geophysics, 44(2): 85-102, effectively quantifies this error as $e = R_n – R_r$, where the data noise, $e$, is determined from the normal and reciprocal resistivity measurement, $R_n$ and $R_r$, respectively. Reciprocal data could have been collected in this study if we had used a dipole-dipole array; however, we found that the measured potentials (signal) associated with the dipole-dipole array were too low at the site (specifically at the pool), so these data sets were abandoned early on in the study.

We have incorporated a comment regarding our limited evaluation of measurement errors (given our choice of measurement array) and cited the work by Slater et al. (2000) on Lines 701-702.

Line 264: "moderate to high damping" – Does this mean different damping factors were used on each dataset? What is the numerical value of damping used and how is this value incorporated into your inversion scheme?

Response: We have revised the text to better reflect our approach Lines 637-648. Moderate dampening was used at the riffle while slightly higher-dampening was applied at the pool. Essentially, the dampening factors are used to reduce the likelihood of non-realistic model parameters through the inversion. This is applied in Res2DInv via the Marquardt-Levenberg modification to the Gauss-Newton equation given by $(\mathbf{J}^T \mathbf{J} + \lambda \mathbf{I}) \Delta \mathbf{q}_k = \mathbf{J}^T \mathbf{g}$, where $\mathbf{J}$ is the Jacabian matrix, $\Delta \mathbf{q}$ is the model parameter change vector, $\mathbf{g}$ is the discrepancy vector between measured and modeled data, and $\lambda$ is the applied dampening factor, ranging from 0.01-1. We found that model solutions were at time converging toward unrealistic resistivities especially during the winter periods. Here, dampening factors of 0.2 and 0.3 for the riffle and pool effectively constrained the range of values of the components of the parameter change vector. Further explanation on this can be found in Loke (2002). We have added a supporting reference in the text on Line 645.

Line 267: What parameters, how were they optimized, and were identical settings used for all datasets?

Response: Our approach was effectively trial by error as is the case of most geophysical inversions. As described in the text, parameters were adjusted in an attempt to reduce the RMS error while maintaining realistic model parameters (i.e., resistivity range). We have revised the text between Lines 637-648 to clarify our approach.

Line 268 – 269: Certainly achieving the lowest possible RMS is not the optimal approach to achieve the most "believable" geophysical result. How does the RMS relate to observed measurement noise/errors? At what point is the inversion fitting noise?

Response: Minimizing the RMS error was done while monitoring the effective range in model resistivity. As we described above we did not collect the data necessary to generate a useful measurement error model based on reciprocal data, therefore, we cannot compare the magnitude of the model RMS relative to a data error model. We acknowledge that this may be a limitation of our study.

Line 280: Is there a reference for this Resistivity Index? What is the justification for manipulating the data in this way?

Response: The resistivity index (a broadly and routinely used normalization technique) was used here so that we could compare the transience observed between the pool and riffle sections relative to a standard datum. One of the reasons for doing this was to remove the biases of varying model range between the pool and riffle. The index simply allowed us to normalize the data sets to their mean value, permitting easier comparison of the timing and magnitude of transient events at each location. We do not believe that this requires a citation since the equation was developed based on the data collection style. We revised the text to better explain the use of this equation (Lines 649-652).

Line 293 – 294: Where is the data demonstrating upward head shown?

Response: This data was not presented and was based on data points not shown in Figure 2. Based on the points monitored in this study within the river (RSG4) and behind the liner (SCV6) the regional groundwater gradients are actually downward, although we do not believe this to be reflective of conditions proximal to the streambed (based on associated datasets). That being said, we have revised the text to reflect the data points that were monitored. These changes are shown between Lines 710-711.

Line 307/Figure 6b: It would be helpful to grade the colors of the lines linearly to more easily show the temperature trend. Even better would be to present these data as a matrix/grid where time is on the x-axis, depth is on the y-axis, and color represents temperature.

Response: We believe the current presentation of the data is reasonable as it is more easily interpreted. The purpose of the figure was to show the temperature dynamics with respect to depth, and the extent of the heterothermic zone, and illustrate the temporal variability overserved during the winter months. We think the current figure layout achieves these objectives.

Line 308: "correspond to areas" I cannot tell from the figure how the fracture patterns correspond to the temperature results. Perhaps some annotation, or another approach to presenting these data would help.

Response: In this case fractures are readily evident along the profile and the deviations in temperature are abundant. There is no singular feature or package of fractures, but rather varying degrees of variability due changing fracture density. This figure conveys the fracture density of sedimentary rock, showing both horizontal and vertical fractures, and sensitivity of the temperature profile in relation to the fracture distribution. This is meant to be first-order view of the fractures and thermal variably beneath the riverbed, so no additional annotation have been added.

Line 321 – 331: [Figure 8] This seems more like a discussion point rather than a result.

Response: In some ways this can be viewed as a point of discussion based on measured data; however, we felt that is was best placed prior to the introduction of the resistivity profiles. The figure is based on the results of Figure 5 and 6. We have revised the text (Lines 737-769) and the caption (Lines 1576-1583). We believe the current placement of this information is reasonable.

Line 356 – 358: "greater number of measurements. . ." why would the number of removed data cause higher RMS? Presumably if the data were removed, they would no longer be included in the RMS calculation.

Response: We have revised the text between Lines 795-846 to clarify this point. Essentially, more data points were removed during noisy periods. While bad (failed) measurements were removed, the overall dataset remained noisy overall, thereby resulting in higher RMS errors.

Line 374: What are the observed thicknesses of basal ice and floating ice?

Response: Unfortunately we were not able to measure the thickness of basal ice, but it was visually observed in the field. We have revised the text throughout to clarify this point.

Line 415-416: "groundwater discharge in this section" I don't follow the logic why the relationship between substrate resistivity and "surface water response" indicates magnitude of discharge that could be interpreted in this way. Also, correlation is not demonstrated or quantified.

Response: We have revised the text to simply state that groundwater-surface water mixing either does not occur or that discharge is strong, thereby limiting potential mixing (Lines 917-920). The word correlation has been removed.

Line 432: "strong upward hydraulic gradients" please indicate where this is demonstrated by data.

Response: We have revised/reduced our emphasis of strong upward gradients in light of the reviewer comments above. We recognize that this was conjecture.

Line 436: "likely dominated the bulk electrical response" Why 'likely'? Based on the evidence shown, temp is clearly dominating the ERT signal.

Response: We have now emphasized the importance of temperature on the resistivity signal and removed the word "likely" on Line 965.

Line 445: Where is ground frost or riverbed ice formation measured data shown?

Response: Neither ground frost or river bed ice were explicitly measured in this study. Ground frost was interpreted based on the resistivity data (Figure 11e), while riverbed ice (basal ice) was observed in the field as described in the text.

Line 459 – 473: As previously stated, data showing the frost and ice should be shown.

Response: Please see response above.

Line 474: I'm not sure what evidence directly supports this statement. The provided sensitivity analysis appears to only vary the river water electrical properties; this doesn't seem to directly simulate the presence of ice as claimed in this statement.

Response: We've revised the text on Lines 1029-1040 to address this concern.

Line 479 – 480: Quantify this? Why would inputting a one-half of true river water resistivity lead to "substantial overestimates of river resistivity" – wouldn't the river water resistivity be fixed so that the output = the input?

Response: We were referring to the bedrock resistivity. The surface water resistivity was fixed as stated. We have revised the text on Lines 1029-1040.

Line 486: What about a synthetic model example?

Response: At this point we have limited our results and discussion to the field measurements. A synthetic study would be very informative and could be considered in future work but would require additional borehole information to build a realistic physical model of the fractured bedrock.

Line 510: How does geoelectrical transience translate into hydrological processes?

Response: Changes in electrical resistivity across a fixed area over time are the result of variations in the electrical properties of the pore water. Changes can occur from temperature, specific conductance, or saturation (phase transformation such as ice formation). Surface water and groundwater typically exhibit distinct electrical properties as we show Figure 7. Whether these property variations can be exploited in a bedrock environment using surface geophysical methods has not previously been explored. Our study aims to assess the utility of surface geophysics, while also examining the utility of temperature and EC fluctuations to infer hydrological processes (e.g., thermal conduction, groundwater-surface water exchange, and fracture connectivity) in a bedrock river of varying morphologic conditions. There is an extensive body of literature demonstrating the use of temperature as a tracer for groundwater flux or flow across a riverbed. Here, we explore the sensitivity of surface electrical methods to seasonal temperature transience in the bedrock. This study does not address all elements of the conceptual model, but rather, represents a step toward a robust understanding.

Line 511 – 452: The conclusions section contains substantial summary and could be reworked for improved focus.

Response: The conclusions have been re-written to better highlight the specific contributions of our study and the implications to future work.

Figure 5: The purpose of this figure is unclear and I suggest that it could be deleted. The A/B/C/D locations are already indicated on Figure 12; the river stage information is presented on Figure 4; the location of the model block midpoints does not appear to be substantially important to the manuscript.

Response: Figure 5 has been removed.

Figure 9: Perhaps showing only the difference between these two maps would make interpretation easier? If not only difference, then perhaps just adding a third difference panel. Also, isocontour labels are too small to read.

Response: The revised Figure 8 now includes a % change plot between low and high stage periods, and the isocontours in these plots have been removed for clarity.

Figure 10: What is the model error relative to the measurement errors? What is the purpose of showing these vast bulk averages when that eliminates any of the valuable spatial information yielded by using tomographic methods? Figure 12 seems to be much more useful than this.

Response: The value of Figure 9 is that it documents the full time-series of the resistivity measurements at each location in the river and shows the range in values observed through the seasonal transitions. It also provides information about the ground conditions which have implications to the interpretation of information in the subsequent figures. The vastly different resistivity conditions and their seasonal behavior is more clearly presented in Figure 9, than that (achievable) in Figure 10 or 11. It is also not possible to present all of the data in Figure 10 (2D sections), yet is important that the reader has an appreciation for the temporal position of these snapshots; in other works Figure 9 provides an important point of reference. Figure 9 also provides an overview of the data quality (e.g., data points removed and model RMS error) and the timing of those events. We believe this provides critical context for the transience presented in Figure 11.

It is reasonable to assume that the model RMS error is a representation of the measurement error. However, the source of the error remains unknown. Unfortunately, the nature of the measurement error cannot be examined in this study due to our choice of resistivity array geometry (Wenner vs. dipole-dipole). We believe the Wenner array was the best choice given the site conditions, and thus, are confident in our methodology given what we knew at the time. We believe that future work should consider the source of measurement errors, and the impacts these have on the model.

At this point we are confident that Figure 9 provides important and useful information for understanding the subsequent information of Figure 10 and 11.

Figure 11: Very nice layout and presentation of this figure, however certainly this needs to be replotted to show the difference between (b) through (h) relative to (a) in both columns

Response: Converting resistivity to relative % change was considered for Figure 10, but we believe it is best to maintain the absolute resistivity in this situation for two reasons: 1) the reader will be able compare the actual resistivity between the pool and riffle sections over the seasonal period, and 2) it maintains the structural distribution of the resistivity that would otherwise be lost or de-emphasized. We do evaluate the relative % change in resistivity using a Resistivity Index calculation in Figure 11.

[revised manuscript text omitted]

---

## Referee Report (RR1)

**Review, revised version of "Electrical Resistivity Dynamics beneath a Fractured Sedimentary Bedrock Riverbed in Response to Temperature and Groundwater/Surface Water Exchange" by Steelman, et al**.

The authors presented a well-considered revision and addressed the majority of the suggested changes. The study in its current form is better structured and its purpose is clear. While the introduction / background is now able to explain the purpose of the study, its survey design and its expected results, the amount of the data, results and findings are huge and still hinder at some part the narrative and comprehensibility of the study. I meanwhile tend to think, you could omit some of the results, in order to keep the structure plainer. For example, what does the EMI data essentially contribute to the final findings of this study? I like Figure 8 and the pattern in low to high stage periods become impressive visible, but do you need this additional information for your outcome? Not sure, particularly since you came up with similar results in Figure 11. Omitting of EMI could easily be done and will assist to focus on ERT.

In addition, the conclusion still lacks of a short comprehensive and supra-regional message to the scientific community. In relation to the huge data sets the study dealt with, the conclusion is too large and contained too many site specific information but transferrable conclusions. It should be shortened by emphasizing its transferrable findings.

Please find below a few further suggestions.

 Line 457 Reword "some distances away" into 'in a certain distance'

Line 500 – The Archie equation is still incomplete, I would either, mention that it is simplified or add the missing tortuosity factor and the saturation exponent, even if you later on neglected these variables as mentioned "Eq. (1) is considered to be a reasonable approximation for this environment."

Line 508 - The value 1.4 for $m$ seems reasonable, still please strengthen it by adding a reference for this assumption.

Line 1117 - Change "will be" into "is", but the whole sentence is in my opinion not a proper start for a conclusion. The positions of the transects within the riverbed will influence the ERT data anyway, at least spatial, even if no flow dynamic occurred at all. I would reword it because you are not interested on different ERT responses but in its potential to discover flow zones and its dynamic, right? Maybe you should start with something like: "We performed a time-lapse resistivity measurements collected across a 200 m reach of the Eramosa River during low and high-stage periods. The results showed highly spatio-temporal variability within the riverbed morphology, which could be attributed to its exposure of bedding plane, vertical fractures, and competency of the rock surface."

Line 1126 to 1174 These are good findings; still, the amount of lines should be shortened.

Line 1175 - I like the last paragraph, however please reword line 1175 toward your desired outcome, instead of "This study demonstrated that surface electrical resistivity has the capacity to detect and resolve changes in electrical resistivity due to…" I would rather go for somewhat like "This study demonstrated that time-lapse ERT has the capacity to image the magnitude and scale of transience within the riverbed…". Your results are strong enough for a more confident statement.

---

## Author Response (AR2)

**Author Response to Reviewer Comments**
**Hydrol. Earth Syst. Sci. Discuss., doi:10.5194/hess-2016-559, 2016**

Author response is shown in blue:
* * *
**Reviewer 1: General Comments**

The authors presented a well-considered revision and addressed the majority of the suggested changes. The study in its current form is better structured and its purpose is clear. While the introduction / background is now able to explain the purpose of the study, its survey design and its expected results, the amount of the data, results and findings are huge and still hinder at some part the narrative and comprehensibility of the study. I meanwhile tend to think, you could omit some of the results, in order to keep the structure plainer. For example, what does the EMI data essentially contribute to the final findings of this study? I like Figure 8 and the pattern in low to high stage periods become impressive visible, but do you need this additional information for your outcome? Not sure, particularly since you came up with similar results in Figure 11. Omitting of EMI could easily be done and will assist to focus on ERT.

Response: The reviewer raises a valid point. We agree that the paper could be presented or published without the EMI results since the main conclusions primarily based on the ERT data which would shorten and streamline the discussion to some degree. However, after some careful consideration we think that the EMI does provide an important spatial element to the study, and provides supporting information regarding the electrical properties of the rock around the ERT datasets. The ERT design was also based on the EMI datasets, so it seems reasonable for this data to be presented in the paper if it is available.

By including the EMI data we are able to comment on the riverbed morphology and nature of the ERT transience, which would otherwise not be possible. We also think that the reader would be more informed about the site and how the ERT results fit in bigger picture. For instance, the transition from rubble to competent rock (riffle to pool) can be observed in the EMI results. It is also another line of evidence that shows measurable electrical transience within the bedrock.

At this stage we prefer to include the EMI data in our results and discussion.

In addition, the conclusion still lacks of a short comprehensive and supra-regional message to the scientific community. In relation to the huge data sets the study dealt with, the conclusion is too large and contained too many site specific information but transferrable conclusions. It should be shortened by emphasizing its transferrable findings.

Response: We have shorted the conclusions and improved the final paragraph to better highlight the broader applicability of our study to scientific community.

**Specific Comments:**

Line 457 Reword "some distances away" into 'in a certain distance'

Response: We have revised the text as suggested.

Line 500 – The Archie equation is still incomplete, I would either, mention that it is simplified or add the missing tortuosity factor and the saturation exponent, even if you later on neglected these variables as mentioned "Eq. (1) is considered to be a reasonable approximation for this environment."

Response: We have updated the equation to include a saturation component as suggested. Slight changes to the accompanied text were made.

Line 508 - The value 1.4 form seems reasonable, still please strengthen it by adding a reference for this assumption.

Response: We have included a citation.

Line 1117 - Change "will be" into "is", but the whole sentence is in my opinion not a proper start for a conclusion. The positions of the transects within the riverbed will influence the ERT data anyway, at least spatial, even if no flow dynamic occurred at all. I would reword it because you are not interested on different ERT responses but in its potential to discover flow zones and its dynamic, right? Maybe you should start with something like: "We performed a time-lapse resistivity measurements collected across a 200 m reach of the Eramosa River during low and high-stage periods. The results showed highly spatiotemporal variability within the riverbed morphology, which could be attributed to its exposure of bedding plane, vertical fractures, and competency of the rock surface."

Response: We have modified the text based these recommendations.

Line 1126 to 1174 These are good findings; still, the amount of lines should be shortened.\

Response: We have shorted the conclusion section by removing the summary of results.

Line 1175 - I like the last paragraph, however please reword line 1175 toward your desired outcome, instead of "This study demonstrated that surface electrical resistivity has the capacity to detect and resolve changes in electrical resistivity due to…" I would rather go for somewhat like "This study demonstrated that time-lapse ERT has the capacity to image the magnitude and scale of transience within the riverbed…". Your results are strong enough for a more confident statement.

Response: We have modified the sentence based on these recommendations.
* * *
**Reviewer 3: General Comments**

This paper is on the topic of ERT method in the fractured sedimentary bedrock riverbed in response to temperature and groundwater/Surface water exchange. The manuscripts is well written, structured and citied. The interpretation of geophysical data, though qualitative, is likely as well as supported by direct data. Research provides a useful approach to investigation into the river flow regime. I think that probably to have more control over the processes that affect the resistivity should have done more profiles of ERT in pool and riffle. I think the manuscript can be accepted for publication.

Response: No further comments.

[revised manuscript text omitted]